# Electroconvective viscous fingering in a single polyelectrolyte fluid on a charge selective surface

Jeonghwan Kim[1,3], Joonhyeon Kim[1,3], Minyoung Kim[2] & Rhokyun Kwak [1] ✉

When a low-viscosity fluid displaces into a higher-viscosity fluid, the liquid-liquid interface becomes unstable causing finger-like patterns. This viscous fingering instability has been widely observed in nature and engineering systems with two adjoined fluids. Here, we demonstrate a hitherto-unrealizable viscous fingering in a single fluid-solid interface. In a single polyelectrolyte fluid on a charge selective surface, selective ion rejection through the surface initiates i) stepwise ion concentration and viscosity gradient boundaries in the fluid and ii) electroconvective vortices on the surface. As the vortices grow, the viscosity gradient boundary pushes away from the surface, resulting viscous fingering. Comparable to conventional one with two fluids, i) a viscosity ratio ($M$) governs the onset of this electroconvective viscous fingering, and ii) the boundary properties (finger velocity and rheological effects) - represented by $M$, electric Rayleigh ($Ra_E$), Schmidt ($Sc$), and Deborah ($De$) numbers - determine finger shapes (straight v.s. ramified, the onset length of fingering, and relative finger width). With controllable onset and shape, the mechanism of electroconvective viscous fingering offers new possibilities for manipulating ion transport and dendritic instability in electrochemical systems.

Electroconvection (EC) is a form of convective motion that originates from the electrokinetic instability at the interface between an electrolyte and a charged surface, such as an electrode or ion exchange membrane[1–8]. This instability influences the distribution and dynamics of ion transport in electrochemical systems where cations or anions are selectively consumed or generated on the surface under an electric field. Such biased ion transport changes in the ion concentration in an electroneutral bulk electrolyte, giving rise to ion concentration polarization (ICP). ICP encompasses both an ion depletion zone and an ion enrichment zone, resulting from the selective ion transport of ions through ion exchange membranes[9–11]. If an applied voltage exceeds a certain threshold, a space charge layer with imbalanced charge forms from strong selective ion rejection through the surface, leading to hydrodynamic instability and the emergence of vortical flows on the surface, that is, EC.

The dynamics and fluidic structures of EC have been studied for various geometric effects (e.g., 2D–3D dimensionality[5,12] and wall confinement[13]) and operating conditions (e.g., symmetric-to-unidirectional[2] or coherent-to-chaotic transition[1]). Based on scientific studies, engineering applications have been tried to control EC. For instance, EC can enhance ion transports on membranes/electrodes beyond diffusion limitation (a.k.a. overlimiting current), so various strategies are suggested to strengthen EC for better desalination or microfluidic preconcentration[10,12,14]. On the other hand, EC causes non-uniform ion fluxes on the surfaces, resulting in fast dendrite growth on electrodes; therefore, shear flow and/or polymer additives were applied to suppress EC in electrodeposition and battery systems[15,16].

The influence of passive scalars on the dynamics of electroconvective flows has received relatively little attention. If these scalars affect the physical properties of the fluid, such as density or viscosity,

[1]Department of Mechanical Convergence Engineering, Hanyang University, Seoul 04763, Republic of Korea. [2]Department of Chemical Engineering, The Pennsylvania State University, University Park, PA 16802, USA. [3]These authors contributed equally: Jeonghwan Kim, Joonhyeon Kim. ✉e-mail: rhokyun@hanyang.ac.kr

chemo-hydrodynamic instabilities may occur in solutions containing reactive chemicals that alter these properties[17]. While Karatay et al. studied the coupling effects of EC and buoyancy force induced by a density gradient[18], the coupling effects of EC and viscosity gradient have not been considered yet, even though viscosity gradient is an important source of hydrodynamic instabilities[9,19,20]. For example, viscous fingering occurs between the interfaces of two fluids under the condition that a low-viscous fluid displaces a highly viscous one. After addressing that the dynamics of viscous fingering is governed by viscosity ratio and capillary number, subsequence studies considered the effects of interfacial conditions[19–22], viscoelastic/-plastic properties[19,20,23], wettabilities[20], geometries[24], and chemical reactions that modulates fluid viscosity[17,25].

The coupling between reactive processes and electrokinetics is a crucial aspect in fields such as electrodeposition and energy storage systems in batteries. These systems experience both EC and viscosity change at the fluid–solid interface as a result of electrochemical reactions[4,8,16,17,25–27]. In this article, we investigate the interplay between EC and viscosity change for the first time, in a single polyelectrolyte fluid on a charge-selective surface. As the polyelectrolyte solution varies in viscosity with its concentration, selective ion rejection through the surface initiates (i) stepwise ion concentration and viscosity gradient boundaries in the fluid and (ii) EC vortices on the surface. If the viscosity gradient is in the same direction as the direction of EC growth, a hitherto-unrealizable viscous fingering can occur on a single fluid–solid interface, which is surprising because two adjoined fluids seem inevitable for viscous fingering[19,20,23,28]. If the viscosity gradient and EC growth are in the opposite direction, EC is strongly suppressed. In addition to this onset dynamics of electroconvective viscous fingering, new scaling relations are established for finger shapes (straight v.s. ramified, the onset length of fingering[29], and relative finger width).

## Results

Our visualization platform to study two-dimensional EC with viscosity variations consists of two juxtaposed identical cation exchange membranes (CEMs) (see "Methods" and Supplementary Fig. 1). Three fluids were loaded in a 0.2 mm-depth channel between the membranes. Based on (i) 10 mM NaCl solution as an ordinary electrolyte, (ii) anionic polyacrylic acid (PAA) solution (0–2.0 wt%) or (iii) cationic polyquaternium-10 (PQ-10) solution (0.5 wt%) was added. These polyelectrolytes can vary fluid viscosity with viscoplasticity (Supplementary Note 1 and Supplementary Tables 1 and 2). An anionic fluorescent dye or pH indicator was added to visualize ICP and EC on the membranes; ion depletion zone can be observed as a dark (or white) region caused by the depletion of fluorescent dyes (or pH indicator)[30,31]. Also, EC structure was identified by tracking fluorescent particles (Supplementary Fig. 2).

When an electric field is applied, only cations can pass through the membranes toward the cathode, and anions are dragged toward the anode (but not pass the membrane). Therefore, the ion depletion zone is generated on the anodic side of CEM, and the ion enrichment zone is generated on the cathodic side[9]. In ordinary electrolytes, if a sufficiently strong voltage is applied (>1 V), EC is generated on the membrane, resulting in the circular ion depletion zone with a flat concentration profile as vortices mix the fluid (Fig. 1a, b)[1,32]. In this depletion zone, one pair of symmetric vortices exists, and these vortices grow and push the ion concentration gradient boundary over time (Fig. 1b)[33]. Here, fluid viscosity is constant everywhere, so there is no viscosity gradient but only ion concentration gradient (inset graph of Fig. 1a).

If we add the polyelectrolyte that can change the local fluid viscosity, the viscosity gradient appears in two directions (Fig. 1c, f). Like cations/anions moving in the electrolyte, polyelectrolytes also migrate under the electric field according to their charge, while they cannot

pass through the membrane due to their large sizes. In the case of anionic PAA, they move toward the anode, so its concentration and viscosity decrease on the anodic side of the membrane, where the ion depletion zone occurs (Fig. 1c). Here, the electroneutrality enforces that $Na^+/Cl^-$ concentration, PAA concentration, and viscosity gradient profiles are overlapped. With very low ion concentration in EC (<1/100 of bulk concentration[34]), the viscosity where EC exists can be assumed to be close to that of water. Consequently, as EC vortices grow in the low-viscous depletion zone, they push the viscosity gradient boundary into the high-viscous bulk electrolyte. In this scenario that reminisces viscous fingering, we observe unique finger-like EC patterns (Fig. 1d, e). Its flow structure in the fingers is also similar to the conventional viscous fingering[19,20,22,28] (Supplementary Fig. 2). This electroconvective viscous fingering is a remarkable type of EC-viscous fingering-coupled instability, which is a clear departure from both previous EC in ordinary fluids and conventional viscous fingering that requires two adjoined fluids. Conversely, the cationic polyelectrolyte (PQ-10) moves toward the cathode and accumulates on the CEM, developing a highly viscous region (Fig. 1f). Therefore, the directions of viscosity gradient and ion concentration gradient become opposite. In this case, the high-viscous region on the CEM resists the fluidic motion, so EC is strongly suppressed (Fig. 1g).

Next, a series of systematic experiments were performed to examine the onset and feature characteristics of electroconvective viscous fingering (Fig. 2). In the experiment, we can observe and categorize three EC morphologies with the anionic polyelectrolytes (PAA, 0–2.0 wt%), i.e., (i) circular EC, (ii) straight finger, and (iii) ramified finger (see Supplementary Video 1). In detail, each morphology states distinctive EC growth shapes. (i) Circular EC shows that smaller circular EC vortices are merged into larger circular ones, which is a phenomenon shown in previous studies of EC in ordinary electrolytes (Fig. 2a, b)[2,5]. Electroconvective viscous fingering shows two distinct patterns as conventional viscous fingering does: (ii) the straight finger shows a tight arrangement of fingers without bifurcation (Fig. 2c, d), and (iii) the ramified finger shows sparsely placed fingers with ramified growth structures (Fig. 2e, f). With the cationic polyelectrolyte (PQ-10, 0.5 wt%), EC was suppressed completely in all voltage conditions.

The shapes of traditional viscous fingering have been characterized through various methods, including roughness of interface[28], fractal dimension[23], and relative finger length to the total distance from the inlet[22]. These methods, while effective in categorizing traditional viscous fingers, cannot distinguish circular EC and straight fingers due to their similar contour shapes. To distinguish three EC morphologies quantitatively, therefore, we define two geometric factors that represent the aspect and area ratios of each EC vortex: $S_1 = L/w$ and $S_2 = A_1/A_2$, where EC vortex width ($w$), length ($L$), the area occupied by each finger ($A_1$) and the area of the bounded rectangle of the finger ($A_2$) (Fig. 2g–i). Figure 3a shows that two dimensionless numbers ($S_1$ and $S_2$) separate the experimental cases distinctly; the blunt shape of the circular EC with relatively low $S_1$ and high $S_2$ (circles in red area, Fig. 3a), the sharp densely packed straight finger with relatively high $S_1$ and $S_2$ (crosses in yellow area, Fig. 3a), and the spare ramified finger with relatively low $S_1$ and $S_2$ (squares in green area, Fig. 3a).

To understand the dynamics of electroconvective viscous fingering, we perform the scaling analysis of EC in the presence of a viscosity gradient. We consider two separated regions, i.e., the ion depletion zone with EC (region (a): white area in Fig. 1c) and the bulk electrolyte region (region (b): red area in Fig. 1c), with two different constant viscosities (the viscosity of the water $\mu_a$ and the viscosity of the bulk electrolyte $\mu_b$). We hypothesize that the onset and pattern selection of electroconvective viscous fingering is governed by (i) the viscosity ratio of region (a) and (b) and (ii) the finger velocity of region (a) into region (b), which is matched with the speed of EC (see Supplementary Table 3 for lists of symbols used in the investigation).

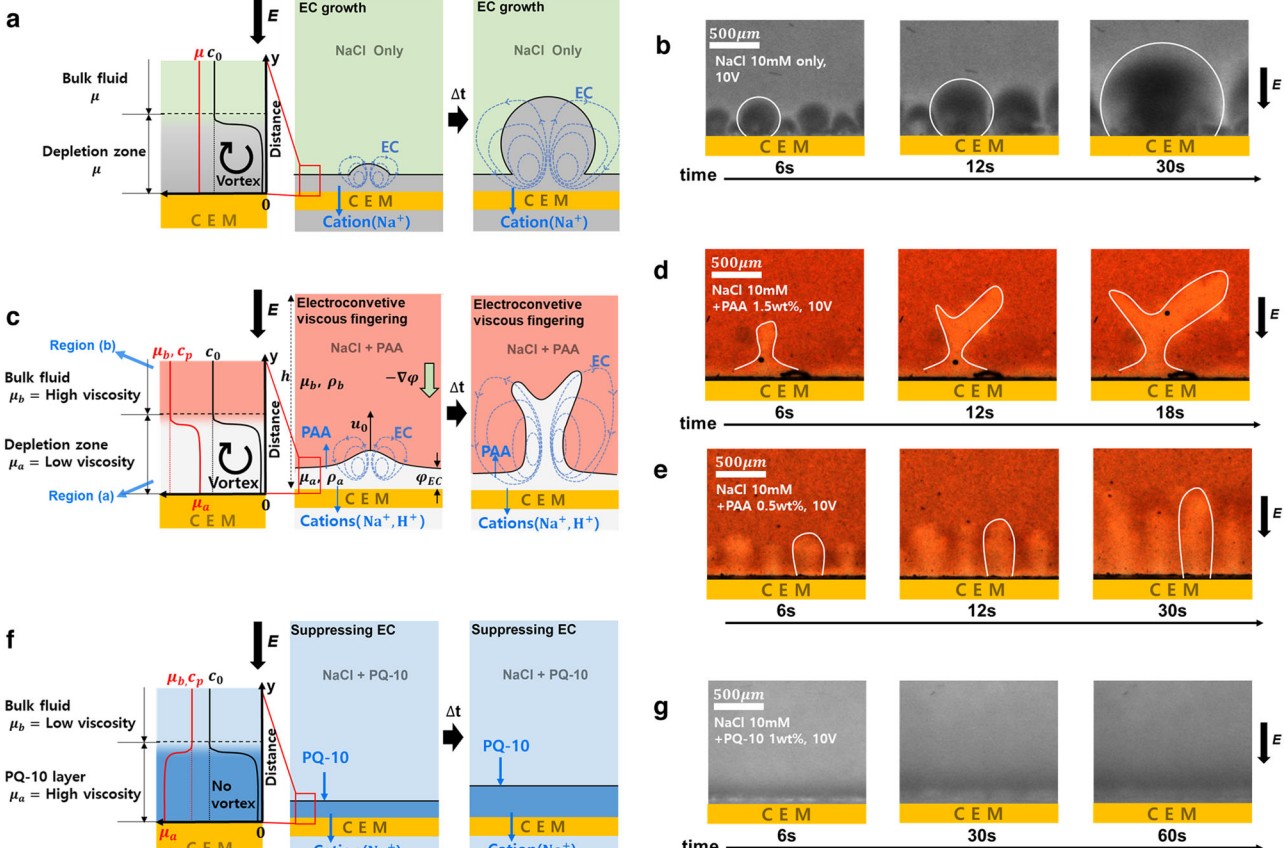

**Fig. 1 | Schematic illustration of EC under a viscosity gradient.** Schematic illustration and experimental images of **a**, **b** circular EC in ordinary electrolytes without viscosity gradient, **c**–**e** electroconvective viscous fingering in an anodic polyelectrolyte (PAA) solution with a viscosity gradient in the $+y$ direction (straight (**d**) and ramified (**e**)), and **f**, **g** suppressed EC in a cathodic polyelectrolyte (PQ-10) solution with a viscosity gradient in $-y$ direction. The depletion zone is exaggerated in scale for intuitive understanding in schematics. Inset graphs in (**a**, **c**, **f**) are ion concentrations ($c_o$, black lines), polyelectrolyte concentration ($c_p$) and fluid viscosity ($\mu_{a,b}$, red lines) profiles in electrolytes. In all three cases, the ion concentrations show the same profile, which has the depletion zone near the membrane. Experimental images were taken by adding fluorescent dyes (**b**, **g**) or a liquid pH indicator (**d**, **e**) (see "Methods").

First, in region (a), we obtain the scaling value of the finger velocity ($u_0$) which is assumed to be the same as EC vortex velocity ($u_{EC}$). According to refs. 2,35, the momentum equation can be simplified with two terms (viscous and electric body force) without external pressure and flows ($0 = \mu_a \boldsymbol{\nabla}^2 \mathbf{U}_a + \varepsilon_a \boldsymbol{\nabla}^2 \varphi_{EC} \boldsymbol{\nabla} \varphi_{EC}$). By balancing these two terms, $u_{EC}$ can be scaled as $\varepsilon_a \varphi_{EC}^2 / \mu_a d_{EC}$, where $\mu$ is the dynamic viscosity, $\mathbf{U}$ is the velocity vector, $\varepsilon$ is the permittivity, $\varphi_{EC}$ is the electrical potential across the ion depletion zone ($\varphi_{EC} = I(R - R_{ohmic})$, where $I$ is current, $R$ is the total resistance, and $R_{ohmic}$ is the Ohmic resistance without overpotentials, see Supplementary Fig. 3), and $d_{EC}$ is the size of EC. The subscript a indicates that the property is of region (a) (so is subscript b in region (b) below). After we match this EC velocity with the finger velocity, $u_0$ also represents the growth rate of the EC interface ($u_{EC} \sim \varepsilon_a \varphi_{EC}^2 / \mu_a d_{EC} \sim u_0 \sim \partial d_{EC}/\partial t$). As a result, $u_0$ can be scaled as (see Supplementary Note 3 for detailed analysis):

$$u_0 \sim \sqrt{\frac{\varepsilon_a \varphi_{EC}^2}{\mu_a t}}. \tag{1}$$

For region (b), we obtain the dimensionless parameters that come from the nondimensionalized momentum equation. In contrast to region (a), an electric body force becomes negligible since the whole region remains electrically neutral, and the unsteady inertia term remains in the equation because the movement of the viscosity gradient boundary by EC is the main source of the flow (the tilde denotes dimensionless variables):

$$x = h\widetilde{x}, \mathbf{U} = u_0 \widetilde{\mathbf{U}}, t = \frac{h}{u_0} \widetilde{t}, \tag{2}$$

$$M_\rho \frac{h}{\delta_{diff}} \frac{\sqrt{Ra_E}}{Sc \cdot M} \left( \frac{\partial \widetilde{\mathbf{U}}}{\partial \widetilde{t}} + \left( \widetilde{\mathbf{U}} \cdot \widetilde{\boldsymbol{\nabla}} \right) \widetilde{\mathbf{U}} \right) = \widetilde{\boldsymbol{\nabla}}^2 \widetilde{\mathbf{U}}, \tag{3}$$

$$\left( M_\rho = \frac{\rho_b}{\rho_a}, Ra_E = \frac{\varepsilon_a \varphi_{EC}^2}{\mu_a D_{eff}}, M = \frac{\mu_b}{\mu_a}, Sc = \frac{\nu_a}{D_{eff}} \right), \tag{4}$$

where $h$ is the channel height, $\delta_{diff}$ is the thickness of the diffusion boundary layer ($= \sqrt{D_{eff}t}$), $D_{eff}$ is the effective diffusivity of charged species ($D_{eff} = 2/(1/D_{Na^+} + 1/D_{Cl^-})$, where $D_{Na^+}$ and $D_{Cl^-}$ are the diffusivities of $Na^+$ and $Cl^-$), $\rho$ is the density, and $\nu$ is the kinematic viscosity. While the convective term $(\widetilde{\mathbf{U}} \cdot \widetilde{\boldsymbol{\nabla}})\widetilde{\mathbf{U}}$ is minor in the low Re regime, time-dependent term $\partial \widetilde{\mathbf{U}}/\partial \widetilde{t}$ is not negligible because the circular/finger-like EC vortices are growing in time.

In the nondimensionalized Navier–Stokes equations (Eq. (3)), there are three dimensionless parameters, $M_\rho$, $h/\delta_{diff}$, and $\sqrt{Ra_E}/(ScM)$. $M_\rho$ indicates the density ratio of the two regions, and it can be neglected since the density does not change significantly by adding PAA to the solution ($M_\rho \sim 1$). The effect of $h/\delta_{diff}$, which indicates the thickness of the diffusion boundary layer, is also negligible

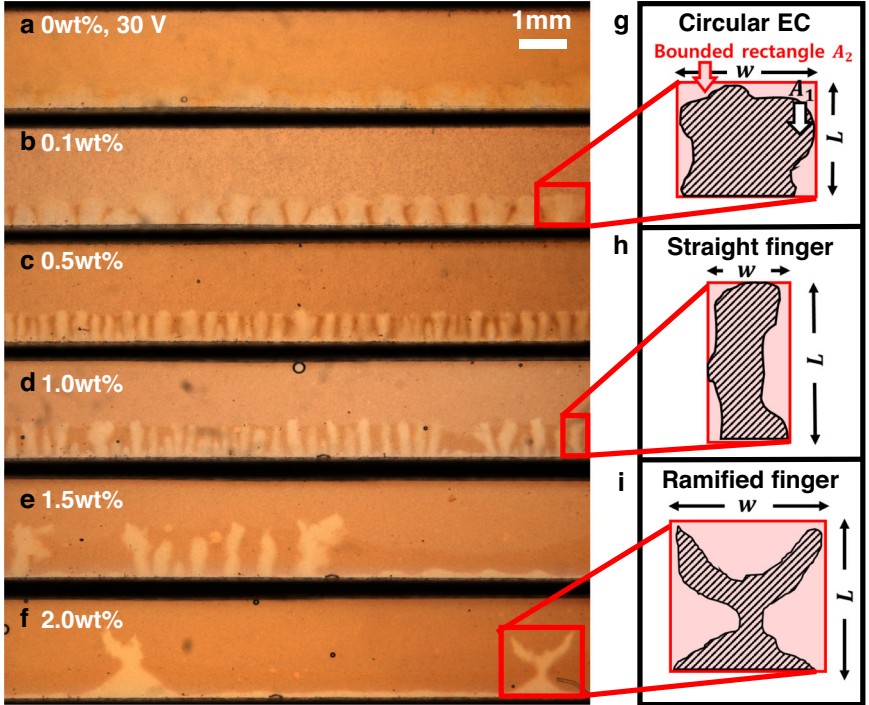

**Fig. 2 | Microscopic visualization of EC and electroconvective viscous fingering.**
**a**, **b** The circular EC is observed at the applied voltage of 30 V in 0–0.1 wt% of the PAA solution; chaotic EC is observed in 0 wt%, which became relatively stable in 0.1 wt%. Above 0.5 wt%, the electroconvective viscous fingering occurs with **c**, **d** straight or **e**, **f** ramified fingers. The main channel between CEMs was filmed above the device. As the liquid pH indicator is used for visualization, the PAA solution maintains its acidic index of red. The depletion zone without the PAA and the pH indicator appears to be white. **g–i** Examples of EC morphologies with the width $w$, length $L$, the finger area $A_1$, and the area of the bounded rectangle $A_2$. In the experiments of 0.5 wt% and 1.0 wt%, where the fingers appear densely, the PAA accumulates between the fingers as the inflow of ECs to the membrane makes a hotspot (see Supplementary Fig. 4). The hotspot becomes more viscous, which inhibits flow but does not inhibit flow where the finger grows. The particle tracking images in Supplementary Fig. 2 show that the finger continues to grow even when the flow is inhibited in the hotspots. In addition, the pH change due to water splitting on the membrane is also negligible since EC plays a dominant role in the overlimiting transport of ions in dilute solutions, and the degree of water splitting is minimal[41–43]. It is demonstrated by the pH profile in Supplementary Fig. 4: the abnormal pH increase (or decrease) is not observed on the anodic (or cathodic) side of the CEM as OH⁻ (or H⁺) is released. The video is available online (Supplementary Video 1).

because this layer should be fully developed to touch a nearly zero ion concentration at the electrolyte-solid interface for initiating EC[34]. In $\sqrt{Ra_E}/(ScM)$, the electric Rayleigh number ($Ra_E$) is the ratio between the electric body force and the viscous force in region (a), which is known to determine the occurrence of EC; EC is generated over a threshold $Ra_E$ as the driving electric body force overwhelms the viscous friction. The Schmidt number ($Sc$) is the ratio between the momentum diffusion and the mass (i.e., ions) diffusion also in region (a). The viscosity ratio ($M$) between the two regions is generally used to predict the onset of viscous fingering[21,22,28]. Compared with the non-dimensional electric body force described in ref. 18, $Ra_E^{0.5}Sc^{-0.5}$, the viscosity gradient effect ($M$) is successfully added through this scaling.

It is noted that there are spatiotemporal variations of pH and PAA concentration in the ion enrichment/depletion zones during the developing EC fingers (Fig. 2 and Supplementary Fig. 4). While this variation may induce the changes of local rheological properties, the scaling analysis would be still valid because it does not affect to the boundary of the ion depletion zone and the bulk electrolyte when EC determine its shape (see Supplementary Note 4).

Based on the scaling analysis, the experimental data are mapped on the $Ra_E - M$ plane with four morphological forms: (i) circular EC vortex, (ii) straight finger, (iii) ramified finger, and (iv) no EC (Fig. 3b). The four boundaries between the four morphological regimes can be identified by i) the critical $Ra_E^*$ and $M^*$ for the occurrence of EC, ii) the critical $M^{**}$ for the onset of electroconvective viscous fingering, and iii) the critical $\sqrt{Ra_E}/(ScM)^*$ for dividing the straight v.s. ramified fingers. First, as addressed above, EC will occur if the electric body force overwhelms the fluid viscosity ($Ra_E > Ra_E^* \sim 10^4$). In addition, if the

viscosity gradient comes to mind, $M$ should be larger than 1, which means that the high viscosity near the membrane inhibits EC generation. It is noted that if we add polymer additives that increase the viscosity in a whole fluid, it would simply shift up the critical $Ra_E^*$. However, with the cationic polyelectrolyte, the local fluid viscosity near the membrane increases more at a higher voltage, so EC is clearly suppressed even at high $Ra_E$.

Second, after EC is initiated, the electroconvective viscous fingering occurs at $M > 1.5$, whereas the conventional circular EC occurs at $1 < M < 1.5$. Interestingly, this critical $M^{**}$ value is matched with the critical viscous ratio of the conventional viscous fingering with two miscible adjoined fluids[22]. This result is reasonable that the two regions of the electroconvective viscous fingering ((a) and (b) in Fig. 1c) are miscible too. Third, the criterion that separates the two finger types is the inertia term $\sqrt{Ra_E}/(ScM)^*$ ( = 0.0047) from the governing Eq. (3). According to the analogy of the conventional viscous fingering[19,20,22,28], this indicates that the relatively large inertia is required to propagate fingers without tip splitting (i.e., straight finger). On the other hand, with relatively small inertia, fingers are hampered by viscous force, and it leads the fingers to be bifurcated.

To confirm the generality of electroconvective viscous fingering, we also investigated EC on both AEM and CEM with different types of polyelectrolytes, including anionic/cationic/neutral and/or strong/weak polyelectrolytes (see Supplementary Fig. 5). As can be expected, the polyelectrolytes suppress the EC on the side where they are concentrated under the electric field, while the neutral polymer does not suppress EC at a moderate concentration (1 wt%). Interestingly, through this investigation, we can find one additional condition to

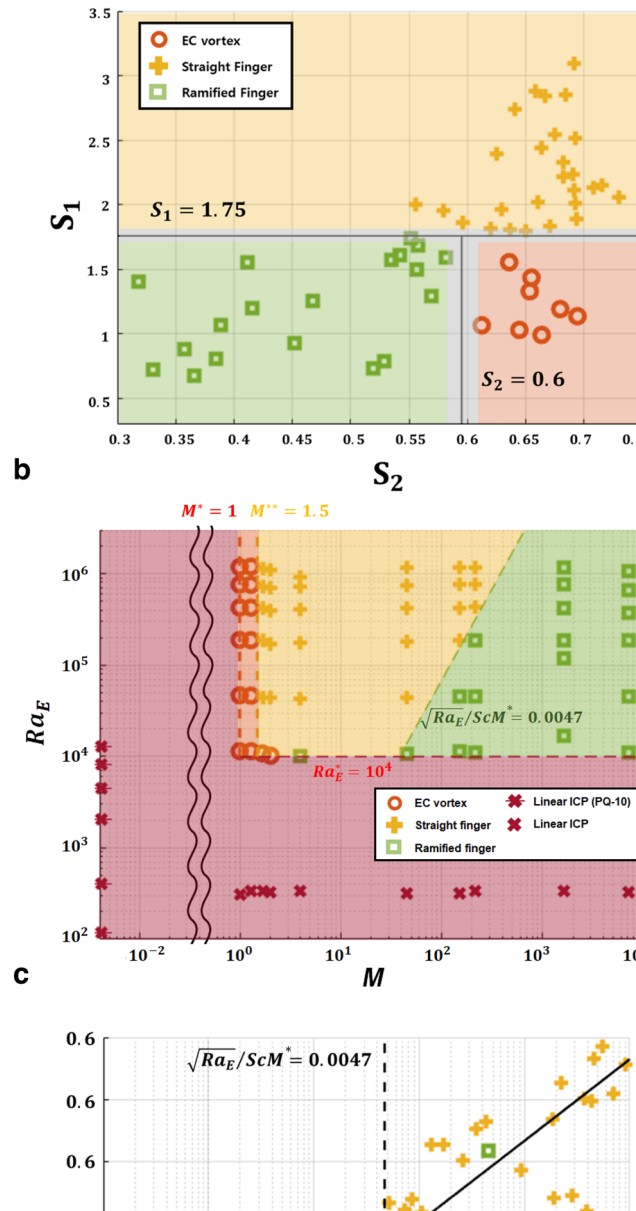

**Fig. 3 | Phase map of the electroconvective viscous fingering and its relative finger width. a** Differentiation of three EC morphologies (circular EC, straight, and ramified finger) by $S_1$ and $S_2$. **b**, Phase map with four regimes as (i) circular EC (orange), (ii) straight finger (yellow), (iii) ramified finger (green), and (iv) no EC (red). The experimental cases cover the ranges of $Ra_E$ (186–6.91×10⁶), $Sc$ (555–3520), and $M$ (0.0041–7.12×10³), as adjusting the applied voltage (1–50 V) and the polyelectrolyte concentrations (0–2.0 wt% PAA and 0.5 wt% PQ-10). Boundaries of each regime are drawn on dotted lines ($M^* = 1$, $M^{**} = 1.5$, $Ra_E^* = 10^4$, and $\sqrt{Ra_E}/ScM = 0.0047$). **c** The relative finger width $\lambda$ according to $\sqrt{Ra_E}/ScM$. The best fitting line is $\lambda = 52.56(\sqrt{Ra_E}/ScM)^{-0.0011} - 51.9563 (R^2 = 0.88)$.

generate the electroconvective viscous fingering, i.e., the viscosity increase effect of a polyelectrolyte should be superior to the conductance increase effect according to its concentration. If not, before we satisfy the requirement condition of EC fingering ($M>1.5$), the solution becomes too conductive to generate strong ion depletion zone and EC (see details in Supplementary Note 2).

For viscous fingering, the relative finger width ($\lambda$) is one of the representative properties to identify the degree of instability[20]. Here, we calculate the relative finger width of the electroconvective viscous fingering as the ratio of the area occupied by fingers to the total area (the tallest finger length ($L$) × the channel width)[36]. Figure 4c shows the correlation between $\lambda$ and $\sqrt{Ra_E}/(ScM)$. The physical meaning of this relation is that viscous force ($ScM$) makes the viscosity gradient boundary stable, while the faster finger movement ($\sqrt{Ra_E}$) make it unstable. Interestingly, our result has the opposite trend that for conventional viscous fingering[19,20]. While conventional one becomes a more stable when a fluid-fluid interface remains in equilibrium by a strong surface tension (showing a large $\lambda$), electroconvective viscous fingering becomes more stable when the number of fingers decreases (showing a small $\lambda$). This difference originates from the source of the fingerings; conventional viscous fingering is generated from the movement of the fluid-fluid interface, but electroconvective viscous fingering stems from EC below the interface. A stronger and more number of EC vortices induces more fingers. In this work, we did not investigate the relationship between the shape factors ($S_1$ and $S_2$) and control parameters. A further comprehensive study is essential for clarifying detailed morphological properties of EC fingers including wavelength, aspect ratio, and fractal dimension to elucidate this relationship.

Next, we investigate the effects of the fluid elasticity on the electroconvective viscous fingering in aspects of EC and viscous fingering (Fig. 4). According to ref. 4, the fluid elasticity does not affect to the onset of EC, but it has two effects on EC: (i) it reduces EC vortex strength and corresponding electroconvective ion transport, resulting a lower overlimiting conductance in current–voltage responses, and (ii) it destabilizes steady EC and promotes faster transition to chaotic EC at relatively low voltage. To investigate these effects of the fluid elasticity in our experiment, we measured the current–voltage responses and the current-time responses in various PAA concentrations (0–2.0 wt%) (Supplementary Figs. 6 and 7). In current–voltage responses with three representative current regimes (Ohmic, limiting, and overlimiting), we confirmed that the critical onset voltage of EC initiating the overlimiting regime remains steady at ~3–4 V, and the ratio of the overlimiting/Ohmic conductance is also kept nearly constant at every PAA concentrations, regardless of whether EC shape is circular or straight/ramified fingers (Supplementary Fig. 6c)[37]. This indicates that the degree of electroconvective ion transport normalized by electromigration is not changed even PAA concentration increases up to 2.0 wt%. As like with conventional EC, the currents also decrease exponentially in time as strong ion depletion zone and EC are developed[37].

The effects of fluid elasticity on viscous fingering is that the normal stress stabilizes the fingering instability, resulting in (i) delaying the onset of viscous fingering, (ii) reducing the number of fingers, and (iii) increasing the width of each finger[38]. The normal stress tends to restore stretched/compressed polymer chains in the flow field and make the fingers move back to the opposite direction of flow. To identify the elasticity effects, we measured the first normal stress difference ($N_1$) of 0.1–2.0 wt% PAA solutions (Supplementary Fig. 8), and compared it with the tendency of the viscosity ($\mu_b$) and the yield stress ($\tau_y$) changes (Fig. 4a–c). All three properties are considerably increased after its overlap concentration ($c^* = 0.84$ wt%). In our experiments, first, the delayed onset of electroconvective viscous fingering is clearly observed in Supplementary Video 1; while the circular EC (0–0.1 wt% PAA) and the straight EC fingers (0.5–1 wt% PAA) start to

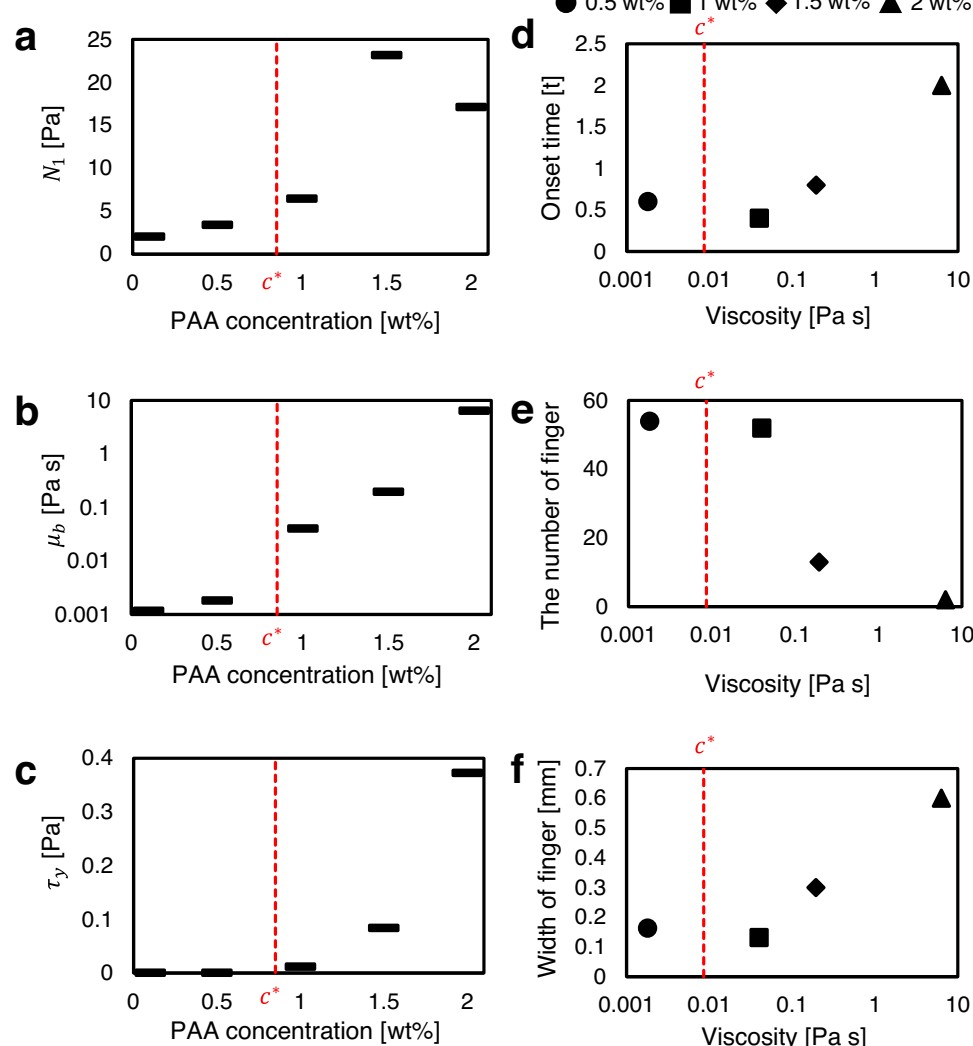

**Fig. 4 | Fluid elasticity effects on electroconvective viscous fingering. a** The first normal stress difference ($N_1$), **b** viscosity ($\mu_b$), and **c**, yield stress ($\tau_y$) v.s. PAA concentrations. $N_1$ was averaged in shear rate range ($<1\,s^{-1}$) because EC experience the shear rate lower than $1\,s^{-1}$ ($N_1$ data are obtained from the rheometer tests in Supplementary Fig. 8). **d** The onset time of EC, **e** the number of fingers and **f** its width were measured from the experiment in Fig. 2 and Supplementary Video 1 (the applied voltage of 30 V in 0.1–2 wt% PAA solutions). $c^* = 0.84$ wt% is the overlap concentration of the PAA solution.

develop right after the voltage applied, the ramified EC fingers (1.5–2 wt% PAA) is observed after considerable time. If we define the time for the EC onset as the time when the color profile shows noticeable change near the membrane, we can figure out that (i) the delayed onset occurs when the fluid elasticity becomes non-negligible as the first normal stress difference ($N_1$) is step-jumped at >1 wt% PAA, and then (ii) the degree of the onset delay is proportional to the viscosity (Fig. 4d). Next, in Fig. 2c–f, we can observe the decrease in the number of straight/ramified EC fingers and the increase of the finger width as PAA concentration increases. The number of the fingers is of course inversely proportional to its width. Interestingly, as like the EC onset delay, the change of the finger number and width are also initiated as $N_1$ is jumped, and the degree of the change is proportional to the viscosity (Fig. 4e, f).

Lastly, in follow up to the elasticity effects, we investigate the influence of yield stress on the formation of the electroconvective viscous fingering. In conventional viscous fingering, Lindner et al. identified that the existence of a yield stress ramifies the fingers at relatively low finger velocities[19]. Indeed, the ramification of the electroconvective viscous fingering is observed only in the fluids with a high concentration of PAA > 0.75 wt% that shows non-negligible yield stresses (Fig. 3b and Supplementary Table 2). Observing this

ramification more closely, the viscosity gradient boundary first deforms as the EC occurs on the membrane ($t = 3$ s, Fig. 5a). Since the bulk electrolyte (i.e., region (b)) is a yield stress fluid, the certain degree of deformation (corresponding high shear stress) is required to penetrate EC into the bulk region and make fingers ($t = 6.2$ s, Fig. 5a). We define the distance when EC penetration occurs as the critical EC length ($d_{EC,critical}$). We then predict $d_{EC,critical}$ by calculating a Deborah number ($De = t_{relax}/t_{process}$) that describes the degree of the solidity and/or fluidity of materials (Fig. 5b). When the time for adjusting to applied stresses or strains (i.e., relaxation time, $t_{relax}$) on the material is greater than the time scale for an experiment (i.e., processing time, $t_{process}$), it is considered as a solid-like material. Here, $t_{relax}$ is the required time to yield the bulk electrolyte (in region (b)), and $t_{process}$ is the exposure time that the viscosity gradient boundary is exposed to EC, resulting (see Supplementary Note 5 for detail derivation):

$$De_{cr} = \frac{t_{relax}}{t_{process}} \sim M^{0.917}\frac{\varphi_{EC}^2}{d_{EC,critical}^2}. \tag{5}$$

When $De < De_{cr}$, the bulk electrolyte starts to yield on the viscous boundary layer, which allows EC to penetrate into the bulk electrolyte

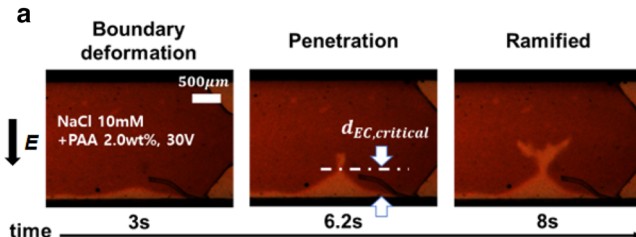

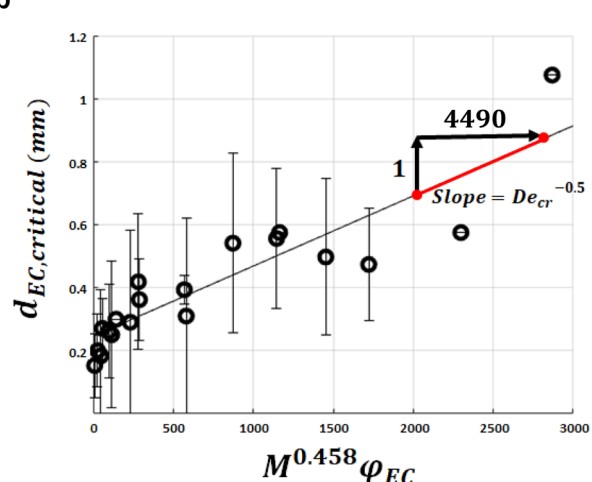

**Fig. 5 | The critical EC length to penetrate a yield stress fluid. a** The generation process of the ramified electroconvective viscous fingering. The video is available online (Supplementary Video 2). **b** The critical EC length ($d_{EC,critical}$) with respect to $M^{0.458}\varphi_{EC}$. The best fitting line is $d_{EC,critical} = 0.0002227 M^{0.458}\varphi_{EC} + 0.2459$ ($R^2 = 0.79$), where the slope of the fitting line indicates $De_{cr}^{-0.5} (= 2.227 \times 10^{-4})$. The error bar represents the standard deviation.

and make fingers. This scaling relation successfully predict $d_{EC,critical}$ in experiments as $\sim M^{0.458}\varphi_{EC}$ with constant $De_{cr} (\sim 2.016 \times 10^7)$ (Fig. 5b, see Supplementary Fig. 9 for how to measure $d_{EC,critical}$).

In conclusion, our study sheds light on a previously unknown phenomenon of electroconvection (EC) under a viscosity gradient, which results in a unique form of viscous fingering at a single fluid–solid interface. This is in contrast to conventional viscous fingering that occurs at the interface of two adjacent fluids. Our scaling analysis, which takes into account the interplay between the viscosity gradient effect and EC velocity ($\sqrt{Ra_E}/(ScM)$), successfully predicts the onset and spread of these fingers. In addition, we reveal that the fluid's viscoplasticity plays a crucial role in determining the onset length of the viscous fingering. This breakthrough expands our knowledge of EC and viscous fingering and offers new avenues for manipulating ion transport and controlling dendritic instabilities in electrochemical applications.

## Methods

To visualize the electroconvective viscous fingering, we used 3D printed polydimethylsiloxane (PDMS) blocks for our device, taking advantage of its transparency and flexibility (see Supplementary Fig. 1). The device consists of two PDMS blocks bonded by oxygen plasma treatment, each with three flow channels and four slots for electrodes and membranes. Carbon electrodes (Fuel Cell Store, Inc., CO., USA) and CEMs (RALEX CMHPES, MEGA Inc., Czech Republic) were slotted to one PDMS block before the plasma bonding. All flow channels have a depth of 200 μm and a length of 2 cm. The main channel in the middle is 2 mm wide. The main channel was filled with 10 mM NaCl solution with/without 0–2.0 wt% polyacrylic acid (PAA) (Carbopol® 940 polymer, Lubrizol, USA) or 0.5 wt% polyquaternium-10 (PQ-10)

(Hydroxyethylcellulose ethoxylate, quaternized, Sigma Aldrich, USA), and the side channels were filled with NaCl 10 mM solution only. The anionic fluorescent dye (1.2 μM Alexa Fluor 488, Invitrogen, CA) or universal pH indicator (pH 1–11) (Hydrion One Drop, Micro Essential Laboratory Inc., USA) was dissolved in the solution to visualize EC. The pH indicator was composed of bromothymol blue, methyl red, and thymol blue[39]. We measured the pH of PAA solutions with respect to the PAA concentration by using the pH indicator and also by using a pH electrode (Orion™ 8103 ROSS™, Thermo Scientific, USA), which revealed that the pH is about 2–3 (ranging from 3.55 for 0.1 wt% PAA to 2.77 for 2.0 wt% PAA). By correlating the color of the pH indicator and the pH data from the electrode, we identified the gray value representing pH (1.5–6) and PAA concentration (0–2.0 wt%) together (Supplementary Fig. 4). The detailed visualization method of EC with pH indicator is described in Supplementary Fig. 10.

Since the shear rate in the experiment is low enough to neglect the shear thinning effect ($<1\,s^{-1}$, see Supplementary Fig. 2), we use the zero-shear viscosity of the polyelectrolyte solutions. The zero-shear viscosity of the PAA solution ($\mu_b$) increases from 0.00089 to 6.344 Pa s as PAA concentration increases from 0 to 2.0 wt% (see Supplementary Table 1). Also, viscoplastic property identified by yield stress occurs above 0.75 wt% (see Supplementary Table 2). In the case of PQ-10, zero-shear viscosity increases from 0.00089 to 0.2162 Pa s as its concentration increases from 0 to 0.5 wt%. In the depletion zone (i.e., region (a) in Fig. 1c), the permittivity in the PAA solution ($\varepsilon_a$) is assumed to be that of water as $6.95 \times 10^{-10}\,C^2 s^2 kg^{-1} m^{-3}$ at 25 °C. For PQ-10 solution with 10 mM NaCl, the permittivity is selected from ref. 40 as $\varepsilon_{PQ-10} = 3.54 \times 10^{-10}\,C^2 s^2 kg^{-1} m^{-3}$. We controlled the applied voltage (1–50 V) using the source meter unit (Keithley 2635, Keithley Instruments, Cleveland, Ohio) and obtained microscopic images with an upright microscope (Axio Zoom.V16, Carl Zeiss, Oberkochen, Germany). Experimental images were analyzed using MATLAB (version R2020b, The Mathworks Inc., Natick, USA), and ImageJ (Rasband, W.S., ImageJ, U.S. National Institutes of Health, Bethesda, MD, USA).

## Data availability

Source data for Figs. 3–5 are provided in the Supplementary Table. All other relevant data supporting the findings of this study are available from the corresponding author on request. Source data are provided with this paper.

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

## Acknowledgements

This work was supported by the Individual Basic Science & Engineering Research Program (NRF-2022R1A2C44001521), the Joint Research (NRF-2021K1A4A7A02102628), and STEAM Programs (NRF-2022M3C1A3090830) from the National Research Foundation of Korea. This research was also supported by the Human Resource Development Program for Industrial Innovation (P0017306) supervised by the Korea Institute for Advancement of Technology.

## Author contributions

R.K. conceived the project, and M.K. conducted the initial feasibility tests. Jeonghwan Kim and Joonhyeon Kim performed the experiments and analyzed the data. Jeonghwan Kim, Joonhyeon Kim, and R.K. performed the theoretical modelings and wrote the manuscript.

## Competing interests

The authors declare no competing interests.
