## [Peer Review File · Nature Communications]

REVIEWER COMMENTS

Reviewer #2 (Remarks to the Author):

The authors present an innovative study analyzing the occurrence of viscous fingering induced by electroconvection. The study encompasses rheological and fluid dynamic aspects coupled with the electroconvection phenomenon. The authors provide a significant amount of data derived from an exhaustive experimental study demonstrating the presence of this phenomenon in the presence of a polyelectrolyte such as PAA. Additionally, a comprehensive dimensional analysis is conducted to characterize the observed instabilities.

While the study is concise, it covers all aspects comprehensively. Undoubtedly, this work opens up new possibilities in the field of non-classical fingering instabilities.

That being said, based on the innovative and exhaustive work carried out, I highly recommend this paper for publication. However, I would like to make some suggestions and ask a few questions:

1) The images in the main text appear pixelated, unlike the clear images in the supplemental information. I'm unsure whether this is due to the compilation process by Nature or if the images provided in the document are compressed. In either case, I recommend re-uploading the images with higher resolution in the final version. Specifically, Figure 1, which schematically explains the dynamics of the system, contains a significant amount of text that is not clearly legible when printed or viewed digitally. The same issue applies to Figures 3 and 4, where the text appears with low resolution.

2) There is a typo in line 190: "... Compared with the nondimensiona9l..."

3) In Supplementary Figure 1, it would be helpful to include a line in Subfigure (a) indicating the cross-section presented in Figure (b). This would aid in better understanding the arrangement of that diagram.

4) The appropriate citations for Matlab and ImageJ should be included (Page 15, line 361).

5) The authors mentioned the use of a pH indicator to observe hydrogen ions (Supplementary Figure 9). Regarding this, could the authors specify the molecule and pH range specifically used as the pH indicator? (The mentioned reference corresponds to a broad kit with several components.) I would also suggest the authors provide estimates of the pH observed in the main experiments. It would be interesting to know at least the pH difference or values between different regions, as the viscosity of PAA changes significantly with pH. Moreover, this is also important as the PAA molecule elongates at basic pH and switches from a globular to a rod-like structure. Consequently, the Radius of Gyration and some of the scalings may be affected as well. Could you provide more discussion on this aspect?

6) One final technical question. It is known that the elastic property of polymer solutions increases considerably above the overlap concentration. Additionally, elasticity may influence and affect viscous fingering patterns. In various studies using PAA solutions, it has been determined that the elasticity of the studied solution is negligible (some authors measured N_1 , the first difference of normal stress, while others conducted an oscillatory analysis). Considering that in this study, PAA concentrations of up to 2% wt were used, and the overlap concentration (C^*) is 0.84 wt%, I would like to inquire: How can elasticity may influence or affect the obtained patterns? Could the authors comment on this aspect?

Beyond these minor issues, I recognize the novelty of this work, and I believe it deserves to be published.

Reviewer #3 (Remarks to the Author):

The paper "Electroconvective Viscous Fingering in a Single Polyelectrolyte Fluid on a Charge Selective Surface" describes interesting experimental findings related to the formation of fluid patterns during electroconvection, when the electrolyte contains poly-electrolytes. The experiments provide a clear picture of the dynamics that take place.

For this, the authors have included 2 different polyelectrolytes, anionic or cationic, in their system and analyzed the dynamics that occur when a strong enough electric field is present, governed by the ion transport through a charge-selective membrane. The relevance for this can be found in electric driven separation processes, as well as in electrochemical conversion processes. As such, I qualify the relevance as very significant.

Overall, I am enthusiastic about the paper, as it provides insight into a topic that, to my understanding, has been overlooked so far. Most electroconvection instability has been related to salt-based electrolytes without polymer additives.

Some comments regarding the experiments: The authors have used PAA as the anionic polyelectrolyte, which is a weak polyelectrolyte. Local pH differences/gradients could thus result in affecting the charge and conformation of the PAA. Such pH effects will result in strong viscosity responses as well, purely based on the polymer configuration (as opposed to its concentration). The polycation is a strong polyelectrolyte. The authors could expand their experimental observations by including a strong anionic polyelectrolyte (e.g. based on sulfonated polymers), and a weak cationic. Also, the inclusion of a neutral polymer could be insightful.

Furthermore, it is known that under strong concentration polarization, water splitting could also occur which gives rise to strong pH gradients.

Regarding the scaling for the b region, I have more doubts:

First, I do not understand the argument for including the inertial term, although the conditions are likely within the low Re regime.

Second, the viscosity is taken outside of the diffusive term, although the viscosity is obviously not constant. Within region b, one expects a gradient in polyelectrolyte concentration, as is also found for ionic electrolytes (see e.g. Influence of Rayleigh-Bénard convection on electrokinetic instability in overlimiting current conditions. *Physical Review Fluids* 2, 033701 (2017)). Although the scaling may work, it is based on arguments that need better support.

Reviewer #4 (Remarks to the Author):

Review of

“Electroconvective Viscous Fingering in a Single Polyelectrolyte Fluid on a Charge Selective Surface”

By Jeonghwan Kim, Joonhyeon Kim, Minyoung Kim and Rhokyun Kwak

This manuscript describes the development 'fingers' in a polyelectrolyte solution adjacent to a charge selective interface under the influence of a potential gradient. In essence, the phenomenon combines electrohydrodynamic effects with local changes in viscosity which in turn induce a phenomenon similar to traditional viscous fingering. Conceptually, this is quite interesting, especially because the electric field drives both the local change in viscosity through the movement of polyelectrolyte as well as the viscous fingering because of the electroconvective flow of lower viscosity fluid into a higher viscosity fluid.

Noteworthy results include the visualization of different finger patterns adjacent to the cation exchange membranes, which are related to relevant dimensionless parameters and plotted clearly. On the other hand, the potential applications of this work are relatively unclear from the manuscript. The formation of dendrites is mentioned, but in the manuscript this is not further clarified, and the formation of dendrites would be in the case of a higher concentration of ions at the charge selective interface, which is the opposite of the effects in the current paper. Furthermore, in the conclusion, authors mention that this phenomenon could be helpful in manipulating ion transport, but ion transport is not significantly characterized in this manuscript. However, the phenomenon is interesting in their own respect, albeit for specific fields, mostly in the realm of charge selective membrane processes. The methods are sound, but could be complemented with more detailed characterization of the electric response of the system, as is normally done in papers regarding these depletion zone phenomena at charge selective interfaces.

In my opinion, the manuscript could be improved by considering the following points:

1. From a physical perspective, as soon as the electric field is switched on, the solution will be prone to local viscosity differences as a result of the PAA or PQ being transported in the field. In combination with the development of electroconvection (EC), this will result in significantly different viscosities in different parts of the system and potentially a constant viscosity gradient in the main channel, going from higher viscosity at the anodic end of the channel to water viscosity at the cathodic end of the channel. Can authors comment on this, and specifically address the effect of the EC on the change in local viscosity. In the current manuscript, the assumption is made that the change in viscosity is solely based on the change in concentration of the polyelectrolyte (which may be warranted), but in order to do so, an assessment of the local shear as a result of EC can be helpful to guarantee that the influence of shear thinning is negligible.

2. The ion transport (e.g. IV-curve) in the system is not characterized in the current manuscript. Are the observations all made in what is known as the overlimiting current regime, as this is where EC is induced? And how does the formation of these finger-like patterns influence the overall current? Is the current dropping as the fingers develop further? From the particle tracking images, it seems that EC is reduced as the fingers develop, which would indicate that the current goes down as no salt is replenished to the CEM. Authors should provide the current vs time data for characteristic experiments (e.g. the experiments shown in Supplementary figure 2, or main manuscript figure 2) so that a conclusion can be drawn on how this phenomenon actually influences ion transport.

3. In follow up to this, on page 10 (line 224) the authors state that more EC will lead to more fingering. This in principle is true, but at the same time this will reduce the EC (e.g. the development of the fingers seems to suppress EC), is there a self-dampening effect beyond which this phenomenon becomes steady?

4. Authors assume the diffusivity of Na^+ and Cl^- to be constant independent of the viscosity of the surrounding fluid, while the diffusivity of PAA is considered to change. Can they show with an order of magnitude analysis that the assumption is reasonable?

5. Can the authors comment on the physical relevance of the boundaries shown in Figure 3a? This figure seems to be of limited relevance in comparison to the other figures in the manuscript. My suggestion would be to further clarify why these boundaries are relevant (e.g. compare with traditional viscous fingering) or to combine these S1 and S2 dimensionless parameters with some influence of the applied electric field or the concentration of the polyelectrolyte solution.

Some minor points for consideration:

1. The cross-references between the manuscript and the supplementary information should be checked. For example, on page 8, line 164, authors refer to Supp Text 3, which should be 2. There were a few more instances like this.

2. On page three, line 42, authors speak of local ion generation and consumption, but ions are not generated nor consumed. They are merely concentrated and depleted because they move in the electric field and are blocked by the ion exchange membrane of similar charge, or are selectively transported through the ion exchange membrane of opposite charge. This should be clarified prior to publication to avoid confusion.

Response to comments from the reviewers

Electroconvective Viscous Fingering in a Single Polyelectrolyte Fluid on a Charge Selective Surface (NCOMMS-23-16231)

by Jeonghwan Kim, Joonhyeon Kim, Minyoung Kim and Rhokyun Kwak

REVIEWER COMMENTS

1. Response to reviewer #2:

Comment #1-1. *The images in the main text appear pixelated, unlike the clear images in the supplemental information. I'm unsure whether this is due to the compilation process by Nature or if the images provided in the document are compressed. In either case, I recommend re-uploading the images with higher resolution in the final version. Specifically, Figure 1, which schematically explains the dynamics of the system, contains a significant amount of text that is not clearly legible when printed or viewed digitally. The same issue applies to Figures 3 and 4, where the text appears with low resolution.*

Response #1-1: We appreciate the reviewer's careful comment. We updated the figures to enhance the clarity.

Comment #1-2. *There is a typo in line 190: "... Compared with the nondimensiona9l..."*

Response #1-2: We appreciate the reviewer's helpful comment. In the revised manuscript, we thoroughly reviewed the manuscript to correct the typos and errors.

Comment #1-3. *In Supplementary Figure 1, it would be helpful to include a line in Subfigure (a) indicating the cross-section presented in Figure (b). This would aid in better understanding the arrangement of that diagram.*

Response #1-3: We appreciate the reviewer's insightful comment. We revised the Subfigure (a) as advised (Fig. R1).

Comment #1-4. *The appropriate citations for Matlab and ImageJ should be included (Page 15, line 361).*

Response #1-4: In the revised manuscript, we provided detailed information about Matlab and ImageJ.

Manuscript page 19, lines 443-445:

Experimental images were analyzed using MATLAB (version R2020b, The Mathworks Inc., Natick, USA), and Image J (Rasband, W.S., ImageJ, U. S. National Institutes of Health, Bethesda, Maryland, USA).

Comment #1-5. *The authors mentioned the use of a pH indicator to observe hydrogen ions (Supplementary Figure 9). Regarding this, could the authors specify the molecule and pH range specifically used as the pH indicator? (The mentioned reference corresponds to a broad kit with several components.) I would also suggest the authors provide estimates of the pH observed in the main experiments. It would be interesting to know at least the pH difference or values between different regions, as the viscosity of PAA changes significantly with pH. Moreover, this is also important as the PAA molecule elongates at basic pH and switches from a globular to a rod-like structure. Consequently, the Radius of Gyration and some of the scalings may be affected as well. Could you provide more discussion on this aspect?*

Response #1-5: We appreciate the reviewer's insightful comment. The observation concerning the pH of the PAA solution during the experiments is indeed significant, given that molecular and fluidic properties such as the radius of gyration and viscosity are sensitive to changes in pH^{1,2}.

First, the components in the pH indicator (Hydrion One Drop, Micro Essential Laboratory Inc., USA) include bromothymol blue, methyl red, and thymol blue³, with a pH reading range from 1 to 11.

Second, we confirmed the pH of PAA solutions with respect to the PAA concentration by using the pH indicator and also by using a pH electrode (Orion™ 8103 ROSS™, Thermo Scientific, USA), which revealed that the pH is about 2-3 (ranging from 3.55 for 0.1wt% PAA to 2.77 for 2.0wt% PAA). By correlating the color of the pH indicator and the pH data from the electrode, we identified the gray value representing pH (1.5-6) and PAA concentration (0-2.0 wt%) together (Fig. R2a). If we describe the local pH and PAA concentration profiles for a representative electroconvective viscous fingering, three regions are observed (Fig. R2b-c): i) diffusive ion enrichment zone on the top CEM with a linear increase of PAA concentration, ii) bulk region where the initial pH/concentration maintained (i.e., region (b) in Fig.1c), and iii) ion depletion zone on the bottom CEM with EC and corresponding flat pH/concentration profiles (i.e., region (a) in Fig.1c). During 8 sec operations at the applied voltage of 30 V in 0.5 wt% PAA solution, the diffusive enrichment zone expands only about 0.2 mm (which is well matched with the theoretical diffusion length, $0.1 \text{ mm} \sim \sqrt{4D_{PAA}t}$); whereas EC fingering grows much faster up to ~1 mm (Fig. R2c-d) (D_{PAA} is the diffusivity of PAA, see Supplementary Table 5). Interestingly, we can observe the EC-induced current hotspot between the fingers where the influx of the vortices occurs (white arrows in Fig. R2a)⁴. At this point, not only the depletion zone is suppressed, but also PAA is concentrated (Fig. R2d). Accordingly, as the viscosity between the fingers increases, downward flows between the fingers are considerably suppressed whereas there are upward flows in the fingers (Fig. R3). Such phenomenon is strengthened when EC are densely packed (e.g., 0.5 wt%, 30 V in Supplementary Video 1).

As described above, there are spatiotemporal variations of pH and PAA concentration in the ion enrichment /depletion zones during the developing EC fingers. However, we thought the scaling analysis was still valid due to the following three reasons. First, the bulk region completely separates the ion enrichment and depletion zones until EC touches the enrichment zone after a considerable time (e.g., > 14 sec at the applied voltage of 30 V in 0.5 wt% PAA solution, Fig. R2c-d); at this merging moment, EC already determine its shape. Also, while the viscosity between EC vortices increases as the PAA concentration increases, this is just one of the consequences of EC after it emerged. EC shape (circular or straight or ramified fingers) is still determined by the viscosity gradient of the ion depletion

zone and the bulk region. In addition, we estimated shear rates from the particle tracking images, and confirmed that it is mostly below 1 s^{-1} (Fig. R3). Referring to Supplementary Fig. 12, the shear-thinning effect is not significant at below 1 s^{-1} shear rate, so we also can neglect the viscosity change by shear thinning of polyelectrolytes induced by EC vortices.

Lastly, the pH variation in whole regions is between 1.5 to 4.5 (Fig. R2c-d). Therefore, we can expect the PAA molecules to be in a globular form and the radius of gyration is nearly constant everywhere¹.

In the revised manuscript, we added the above new analysis and discussions as follows:

Manuscript page 18, line 420 – 429:

The anionic fluorescent dye (1.2 μM Alexa Fluor 488, Invitrogen, CA) or universal pH indicator (pH 1-11) (Hydrion One Drop, Micro Essential Laboratory Inc., USA) was dissolved in the solution to visualize EC. The pH indicator was composed of bromothymol blue, methyl red, and thymol blue³. We measured the pH of PAA solutions with respect to the PAA concentration by using the pH indicator and also by using a pH electrode (Orion™ 8103 ROSS™, Thermo Scientific, USA), which revealed that the pH is about 2-3 (ranging from 3.55 for 0.1wt% PAA to 2.77 for 2.0wt% PAA). By correlating the color of the pH indicator and the pH data from the electrode, we identified the gray value representing pH (1.5-6) and PAA concentration (0-2.0 wt%) together (Supplementary Fig. 4).

Manuscript page 9, line 193 – 197:

It is noted that there are spatiotemporal variations of pH and PAA concentration in the ion enrichment /depletion zones during the developing EC fingers (Fig. 2 and Supplementary Fig. 4). While this variation may induce the changes of local rheological properties, the scaling analysis would be still valid because it does not affect to the boundary of the ion depletion zone and the bulk electrolyte when EC determine its shape (see Supplementary Note 4).

Manuscript page 18, line 431 – 433:

Since the shear rate in the experiment is low enough to neglect the shear thinning effect ($< 1 \text{ s}^{-1}$, see Supplementary Fig. 2), we use the zero-shear viscosity of the polyelectrolyte solutions.

Comment #1-6. It is known that the elastic property of polymer solutions increases considerably above the overlap concentration. Additionally, elasticity may influence and affect viscous fingering patterns. In various studies using PAA solutions, it has been determined that the elasticity of the studied solution is negligible (some authors measured N_1 , the first difference of normal stress, while others conducted an oscillatory analysis). Considering that in this study, PAA concentrations of up to 2% wt were used, and the overlap concentration (C^*) is 0.84 wt%, I would like to inquire: How can elasticity may influence or affect the obtained patterns? Could the authors comment on this aspect?

Response #1-6: We are grateful for the reviewer's perceptive insights. The electroconvective viscous fingering has aspects of both electroconvection (EC) and viscous fingering, so we addressed the influence of elasticity on both EC and viscous fingering in the revised manuscript as follows:

According to Li *et al.*⁵, the fluid elasticity does not affect to the onset of EC, but it has two effects on EC: i) it reduces EC vortex strength and corresponding electroconvective ion transport, resulting an lower overlimiting conductance in current-voltage responses, and ii) it destabilizes steady EC and promotes faster transition to chaotic EC at relatively low voltage. To investigate these effects of the fluid elasticity in our experiment, we measured the current-voltage responses from 0 to 7 V at 0.2 V intervals every 30 s and the current-time responses in various PAA concentrations (0-2.0 wt%) (Fig. R4-5). First, we verified that three representative current regimes (Ohmic, limiting, and overlimiting) are clearly observed (Fig. R4a)⁶. Here, the critical onset voltage of EC initiating the overlimiting regime remains steady at ~3-4 V, regardless of whether EC shape is circular or straight/ramified fingers. As like with conventional EC, the current decreases exponentially in time as strong ion depletion zone and EC are developed⁶.

Next, from the current-voltage responses, we estimated the Ohmic and overlimiting conductance, which represents conductive (a.k.a. electromigration) or electroconvective ion transports, respectively (Fig. R4b). Both Ohmic and overlimiting conductance are keep nearly constant at < 1.0 wt%, but it starts to increase > 1.0 wt%. It is probably because the considerable amount of PAA's counter ions (i.e., H^+) is released as PAA concentration increases. Here, the ratio of these two conductance shows similar value at every PAA concentration (Fig. R4c); this indicates that the degree of electroconvective ion transport normalized by electromigration is not changed even PAA concentration increases up to 2.0 wt%. Moreover, we did not detect chaotic EC in our experimental conditions. Therefore, we can say that the fluid elasticity of the 0-2.0 wt% PAA solutions did not affect on EC in this work.

The effect of fluid elasticity on viscous fingering is described in Jangir *et al.*⁷, who conducted a simulation adjusting the relaxation time of the polymeric solution with a fixed displacement velocity and viscosity ratio. The result shows that the normal stress (i.e., elastic effect) stabilizes the fingering instability, resulting in i) delaying the onset of viscous fingering, ii) reducing the number of fingers, and iii) increasing the width of each finger. They explained these effects that the normal stress tends to restore stretched/compressed polymer chains in the flow field and make the fingers move back to the opposite direction of flow.

To identify the elasticity effects, we measured the first normal stress difference (N_1) of 0.1-2.0 wt% PAA solutions (Fig. R6), and compared it with the tendency of the viscosity (μ_b) and the yield stress (τ_y) changes (Fig. R7a-c). As the reviewer pointed out, all three properties are considerably increases after its overlap concentration ($c^* = 0.84$ wt%). In our experiments, first, the delayed onset of

electroconvective viscous fingering is clearly observed in Supplementary Video 1; while the circular EC (0-0.1 wt% PAA) and the straight EC fingers (0.5-1 wt% PAA) start to develop right after the voltage applied, the ramified EC fingers (1.5-2 wt% PAA) is observed after considerable time. If we define the time for the EC onset as the time when the color profile shows noticeable change near the membrane, we can figure out that i) the delayed onset occurs when the fluid elasticity becomes non-negligible as the first normal stress difference (N_1) is step-jumped at > 1 wt% PAA, and then ii) the degree of the onset delay is proportional to the viscosity (Fig. R7d).

Next, in Fig. 2c-f, we can observe the decrease in the number of straight/ramified EC fingers and the increase of the finger width as PAA concentration increases. The number of the fingers is of course inversely proportional to its width. Interestingly, as like the EC onset delay, the change of the finger number and width are also initiated as N_1 is step-jumped at > 1 wt% PAA, and then the degree of the change is proportional to the viscosity (Fig. R7e-f).

In summary, as like the conventional viscous fingering, we can observe the three effects of the fluid elasticity on the electroconvective viscous fingering, i.e., delaying the onset of EC fingering, widening the finger width, and reducing the number of the fingers. All three effects are initiated as the fluid elasticity of the PAA solutions becomes dominant as N_1 is step-jumped at > 1 wt% PAA, and the degree of the effects is proportional to the viscosity.

We added this new result and discussion in the revised manuscript and supplementary information.

Manuscript page 11, line 249 – 282:

Next, we investigate the effects of the fluid elasticity on the electroconvective viscous fingering in aspects of EC and viscous fingering (Fig. 4). According to Li *et al.*⁵, the fluid elasticity does not affect to the onset of EC, but it has two effects on EC: i) it reduces EC vortex strength and corresponding electroconvective ion transport, resulting a lower overlimiting conductance in current-voltage responses, and ii) it destabilizes steady EC and promotes faster transition to chaotic EC at relatively low voltage. To investigate these effects of the fluid elasticity in our experiment, we measured the current-voltage responses and the current-time responses in various PAA concentrations (0-2.0 wt%) (Supplementary Fig. 6-7). In current-voltage responses with three representative current regimes (Ohmic, limiting, and overlimiting), we confirmed that the critical onset voltage of EC initiating the overlimiting regime remains steady at $\sim 3-4$ V, and the ratio of the overlimiting / Ohmic conductance is also keep nearly constant at every PAA concentrations, regardless of whether EC shape is circular or straight/ramified fingers (Supplementary Fig. 6c)⁶. This indicates that the degree of electroconvective ion transport normalized by electromigration is not changed even PAA concentration increases up to 2.0 wt%. As like with conventional EC, the currents also decrease exponentially in time as strong ion depletion zone and EC are developed⁶.

The effects of fluid elasticity on viscous fingering is that the normal stress stabilizes the fingering instability, resulting in i) delaying the onset of viscous fingering, ii) reducing the number of fingers, and iii) increasing the width of each finger⁷. The normal stress tends to restore stretched/compressed polymer chains in the flow field and make the fingers move back to the opposite direction of flow. To identify the elasticity effects, we measured the first normal stress difference (N_1) of 0.1-2.0 wt% PAA solutions (Supplementary Fig. 8), and compared it with the tendency of the viscosity (μ_b) and the yield stress (τ_y) changes (Fig. 4a-c). All three properties are considerably increase after its overlap

concentration ($c^*=0.84$ wt%). In our experiments, first, the delayed onset of electroconvective viscous fingering is clearly observed in Supplementary Video 1; while the circular EC (0-0.1 wt% PAA) and the straight EC fingers (0.5-1 wt% PAA) start to develop right after the voltage applied, the ramified EC fingers (1.5-2 wt% PAA) is observed after considerable time. If we define the time for the EC onset as the time when the color profile shows noticeable change near the membrane, we can figure out that i) the delayed onset occurs when the fluid elasticity becomes non-negligible as the first normal stress difference (N_1) is step-jumped at > 1 wt% PAA, and then ii) the degree of the onset delay is proportional to the viscosity (Fig. 4d). Next, in Fig. 2c-f, we can observe the decrease in the number of straight/ramified EC fingers and the increase of the finger width as PAA concentration increases. The number of the fingers is of course inversely proportional to its width. Interestingly, as like the EC onset delay, the change of the finger number and width are also initiated as N_1 is jumped, and the degree of the change is proportional to the viscosity (Fig. 4e-f).

2. Response to reviewer #3:

Comment #2-1. The authors have used PAA as the anionic polyelectrolyte, which is a weak polyelectrolyte. Local pH differences/gradients could thus result in affecting the charge and conformation of the PAA. Such pH effects will result in strong viscosity responses as well, purely based on the polymer configuration (as opposed to its concentration).

Response #2-1: We would like to express our gratitude to the reviewer for the careful comment. The point raised regarding the local pH differences of the PAA solution is undoubtedly vital as molecular and fluidic properties such as the charge, conformation, and viscosity are known to be sensitive to pH changes^{1,2}.

First, the components in the pH indicator (Hydrion One Drop, Micro Essential Laboratory Inc., USA) include bromothymol blue, methyl red, and thymol blue³, with a pH reading range from 1 to 11.

Second, we confirmed the pH of PAA solutions with respect to the PAA concentration by using the pH indicator and also by using a pH electrode (Orion™ 8103 ROSS™, Thermo Scientific, USA), which revealed that the pH is about 2-3 (ranging from 3.55 for 0.1wt% PAA to 2.77 for 2.0wt% PAA). By correlating the color of the pH indicator and the pH data from the electrode, we identified the gray value representing pH (1.5-6) and PAA concentration (0-2.0 wt%) together (Fig. R2a). If we describe the local pH and PAA concentration profiles for a representative electroconvective viscous fingering, three regions are observed (Fig. R2b-c): i) diffusive ion enrichment zone on the top CEM with a linear increase of PAA concentration, ii) bulk region where the initial pH/concentration maintained (i.e., region (b) in Fig.1c), and iii) ion depletion zone on the bottom CEM with EC and corresponding flat pH/concentration profiles (i.e., region (a) in Fig.1c). During 8 sec operations at the applied voltage of 30 V in 0.5 wt% PAA solution, the diffusive enrichment zone expands only about 0.2 mm (which is well matched with the theoretical diffusion length, $0.1 \text{ mm} \sim \sqrt{4D_{PAA}t}$); whereas EC fingering grows much faster up to ~1 mm (Fig. R2c-d) (D_{PAA} is the diffusivity of PAA, see Supplementary Table 5). Interestingly, we can observe the EC-induced current hotspot between the fingers where the influx of the vortices occurs (white arrows in Fig. R2a)⁴. At this point, not only the depletion zone is suppressed, but also PAA is concentrated (Fig. R2d). Accordingly, as the viscosity between the fingers increases, downward flows between the fingers are considerably suppressed whereas there are upward flows in the fingers (Fig. R3). Such phenomenon is strengthened when EC are densely packed (e.g., 0.5 wt%, 30 V in Supplementary Video 1).

As described above, there are spatiotemporal variations of pH and PAA concentration in the ion enrichment /depletion zones during the developing EC fingers. However, we thought the scaling analysis was still valid due to the following three reasons. First, the bulk region completely separates the ion enrichment and depletion zones until EC touches the enrichment zone after a considerable time (e.g., > 14 sec at the applied voltage of 30 V in 0.5 wt% PAA solution, Fig. R2c-d); at this merging moment, EC already determine its shape. Also, while the viscosity between EC vortices increases as the PAA concentration increases, this is just one of the consequences of EC after it emerged. EC shape (circular or straight or ramified fingers) is still determined by the viscosity gradient of the ion depletion zone and the bulk region. In addition, we estimated shear rates from the particle tracking images, and confirmed that it is mostly below 1 s^{-1} (Fig. R3). Referring to Supplementary Fig. 12, the shear-thinning

effect is not significant at below 1 s^{-1} shear rate, so we also can neglect the viscosity change by shear thinning of polyelectrolytes induced by EC vortices.

Lastly, the pH variation in whole regions is between 1.5 to 4.5 (Fig. R2c-d). Therefore, we can expect the PAA molecules to be in a globular form and the radius of gyration is nearly constant everywhere¹.

In the revised manuscript, we added the above new analysis and discussions as follows:

Manuscript page 18, line 420 – 429:

The anionic fluorescent dye ($1.2 \mu\text{M}$ Alexa Fluor 488, Invitrogen, CA) or universal pH indicator (pH 1-11) (Hydrion One Drop, Micro Essential Laboratory Inc., USA) was dissolved in the solution to visualize EC. The pH indicator was composed of bromothymol blue, methyl red, and thymol blue³. We measured the pH of PAA solutions with respect to the PAA concentration by using the pH indicator and also by using a pH electrode (Orion™ 8103 ROSS™, Thermo Scientific, USA), which revealed that the pH is about 2-3 (ranging from 3.55 for 0.1wt% PAA to 2.77 for 2.0wt% PAA). By correlating the color of the pH indicator and the pH data from the electrode, we identified the gray value representing pH (1.5-6) and PAA concentration (0-2.0 wt%) together (Supplementary Fig. 4).

Manuscript page 9, line 193 – 197:

It is noted that there are spatiotemporal variations of pH and PAA concentration in the ion enrichment /depletion zones during the developing EC fingers (Fig. 2 and Supplementary Fig. 4). While this variation may induce the changes of local rheological properties, the scaling analysis would be still valid because it does not affect to the boundary of the ion depletion zone and the bulk electrolyte when EC determine its shape (see Supplementary Note 4).

Manuscript page 18, line 431 – 433:

Since the shear rate in the experiment is low enough to neglect the shear thinning effect ($< 1 \text{ s}^{-1}$, see Supplementary Fig. 2), we use the zero-shear viscosity of the polyelectrolyte solutions.

Comment #2-2. The polycation is a strong polyelectrolyte. The authors could expand their experimental observations by including a strong anionic polyelectrolyte (e.g. based on sulfonated polymers), and a weak cationic. Also, the inclusion of a neutral polymer could be insightful.

Response #2-2: We are grateful for the reviewer's observation and agree that expanding experimental investigations to include a strong anionic polyelectrolyte, weak cationic polyelectrolyte, and neutral polymer would enhance our understanding. In our manuscript, we used polyacrylic acid (PAA) as a weak anionic polyelectrolyte and polyquaternium-10 (PQ-10) as a strong cationic polyelectrolyte. In the revised manuscript, we employed three additional additives: i) sodium polystyrene sulfonate (NaPSS) as a strong anionic polyelectrolyte, ii) polyallylamine hydrochloride (PAH) as a weak cationic polyelectrolyte, and iii) polyethylene oxide (PEO) as a neutral polymer. We carried out experiments involving solutions containing the above three polymers with the given weight percent as 1 wt%, as well as a 10mM NaCl solution as a reference (Fig. R8). To investigate the effect of polyelectrolyte migration under an electric field based on a charge sign, we utilized a device that comprises juxtaposed anion exchange membrane (AEM) and cation exchange membrane (CEM). The viscosities of the solutions were measured as 1.48, 1.7, 40, and 216 mPa s for 1wt% NaPSS, PAH, PAA, and 0.5wt% PQ-10 solution, respectively (Fig. R8a).

First, as we already observed in the original manuscript, the EC suppression is shown as the cationic (or anionic) polyelectrolytes are concentrated and the viscosity is increased on CEM (or AEM), regardless of whether they are weak or strong (Fig. R8c). The neutral polymer (1wt % PEO) failed to suppress EC as it exhibited no perceptible movements or accumulation anywhere (Fig. R8b).

Next, as we also addressed in the original manuscript, the unique electroconvective viscous fingering occurs when the viscosity ratio M across the depletion and bulk region (region (a) and (b) in Fig.1b) should be higher than 1.5, regardless of whether they are weak or strong. The finger-like EC pattern can be seen in 1 wt% PAA and 1 wt% PQ-10 solutions, both of which have a sufficiently high viscosity ratio ($M = 40$ and 243 , respectively); whereas the conventional circular EC was observed in 1 wt% PAH and 1 wt% NaPSS solutions, both of which have a marginal viscosity ratio ($M = 1.9$ and 1.66 , respectively).

We may increase PAH / NaPSS concentrations to achieve a higher viscosity ratio. However, this requires significantly high weight of these polyelectrolytes because their viscosity increase effect is relatively small (Fig. R8a). If we increase the concentration to get high bulk viscosity (10 wt% of PAH / NaPSS), the solution's conductance (9.8 / 23.16 mS for PAH / NaPSS 10wt% solution, respectively) is too high to initiate strong ion depletion zone and EC (EC occurs when the concentration of NaCl solution at the membrane surface is sufficiently low for initiating ion depletion zone, and most of EC experiments are conducted under NaCl 0.1 M, where the conductance is 7.3 mS)^{8,9}.

In summary, we can find one additional condition to generate EC fingering, i.e., the viscosity increase effect of a polyelectrolyte should be superior to the conductance increase effect according to its concentration. The charge sign of the polyelectrolyte determines the location of EC fingering or EC suppression, regardless of whether the polyelectrolyte is weak or strong.

We added this additional information in the revised manuscript and Supplementary Fig. 5 and Supplementary Note 2.

Manuscript page 10, line 221 – 231:

To confirm the generality of electroconvective viscous fingering, we also investigated EC on both AEM and CEM with different types of polyelectrolytes, including anionic/cationic/neutral and/or strong/weak polyelectrolytes (see Supplementary Fig. 5). As can be expected, the polyelectrolytes suppress the EC on the side where they are concentrated under the electric field, while the neutral polymer does not suppress EC at a moderate concentration (1 wt%). Interestingly, through this investigation, we can find one additional condition to generate the electroconvective viscous fingering, i.e., the viscosity increase effect of a polyelectrolyte should be superior to the conductance increase effect according to its concentration. If not, before we satisfy the requirement condition of EC fingering ($M > 1.5$), the solution becomes too conductive to generate strong ion depletion zone and EC (see details in Supplementary Note 2).

Comment #2-3. Furthermore, it is known that under strong concentration polarization, water splitting could also occur which gives rise to strong pH gradients.

Response #2-3: As addressed in the previous works⁸, water splitting may occur on the membranes when the ion depletion zone is developed, and the strong voltage is applied on there. It is also known that such water splitting supply additional H^+/OH^- ions to mitigate concentration polarization and inhibit EC generation¹⁰. However, generally on chemically stable CEM, EC plays a dominant role in the overlimiting transport of ions in dilute solutions, and the degree of water splitting is minimal^{9,11,12}.

In addition, as we addressed in response #2-1, we profiled the local pH in whole regions (Fig. R2c-d). If there were significant water splitting on CEM, abnormal pH increase (or decrease) would be observed on the anodic (or cathodic) side of the CEM as OH^- (or H^+) is released.

We addressed this discussion in the revised manuscript.

Manuscript page 22, line 488 – 492:

Additionally, the pH change due to water splitting on the membrane is also negligible since EC plays a dominant role in the overlimiting transport of ions in dilute solutions, and the degree of water splitting is minimal^{9,11,12}. It is demonstrated by the pH profile in Supplementary Fig. 4: the abnormal pH increase (or decrease) is not observed on the anodic (or cathodic) side of the CEM as OH^- (or H^+) is released.

Comment #2-4. *First, I do not understand the argument for including the inertial term, although the conditions are likely within the low Re regime.*

Response #2-4: As you pointed out, it is true that the effect of the convective term $((\tilde{\mathbf{U}} \cdot \tilde{\mathbf{\nabla}})\tilde{\mathbf{U}})$ is minor in the low Re regime in the equation (3) in the manuscript. However, the effect of the time-dependent term $(\partial\tilde{\mathbf{U}}/\partial\tilde{t})$ is not negligible because the circular/finger-like EC vortices are growing in time. Indeed, we demonstrated that the morphological characteristics of the EC fingers are governed by the term $\sqrt{Ra_E}/(ScM)$, referred to as inertia term. We addressed above explanation in the revised manuscript. We also refer the equation (3) in the manuscript below.

Equation 3 in the manuscript (page 9, lines 187):

$$M_\rho \frac{h}{\delta_{diff}} \frac{\sqrt{Ra_E}}{Sc \cdot M} \left(\frac{\partial \tilde{\mathbf{U}}}{\partial \tilde{t}} + (\tilde{\mathbf{U}} \cdot \tilde{\mathbf{\nabla}})\tilde{\mathbf{U}} \right) = \tilde{\mathbf{\nabla}}^2 \tilde{\mathbf{U}} \quad (3)$$

Manuscript page 8, line 177 – 178:

While the convective term $(\tilde{\mathbf{U}} \cdot \tilde{\mathbf{\nabla}})\tilde{\mathbf{U}}$ is minor in the low Re regime, time-dependent term $\partial\tilde{\mathbf{U}}/\partial\tilde{t}$ is not negligible because the circular/finger-like EC vortices are growing in time.

Comment #2-5 *Second, the viscosity is taken outside of the diffusive term, although the viscosity is obviously not constant. Within region b, one expects a gradient in polyelectrolyte concentration, as is also found for ionic electrolytes (see e.g. Influence of Rayleigh-Bénard convection on electrokinetic instability in overlimiting current conditions. Physical Review Fluids 2, 033701 (2017)). Although the scaling may work, it is based on arguments that need better support.*

Response #2-5: Thank you for your comment. As we described in the response #2-1, we identified that the region (b) maintained its initial PAA concentration, pH, and corresponding viscosity, until the ion depletion zone and the ion enrichment zone start to overlap after a consideration time (e.g., > 14 sec at the applied voltage of 30 V in 0.5 wt% PAA solution, Fig. R2). Consequently, when EC determine its shape, we can assume that the region (a) and (b) have constant viscosities. We added this discussion in Supplementary Note 4.

3. Response to reviewer #4:

Comment #3-1. From a physical perspective, as soon as the electric field is switched on, the solution will be prone to local viscosity differences as a result of the PAA or PQ being transported in the field. In combination with the development of electroconvection (EC), this will result in significantly different viscosities in different parts of the system and potentially a constant viscosity gradient in the main channel, going from higher viscosity at the anodic end of the channel to water viscosity at the cathodic end of the channel. Can authors comment on this, and specifically address the effect of the EC on the change in local viscosity. In the current manuscript, the assumption is made that the change in viscosity is solely based on the change in concentration of the polyelectrolyte (which may be warranted), but in order to do so, an assessment of the local shear as a result of EC can be helpful to guarantee that the influence of shear thinning is negligible.

Response #3-1: We appreciate the reviewer's insightful comment. The observation concerning the pH of the PAA solution during the experiments is indeed significant, given that molecular and fluidic properties such as the radius of gyration and viscosity are sensitive to changes in pH^{1,2}.

First, the components in the pH indicator (Hydrion One Drop, Micro Essential Laboratory Inc., USA) include bromothymol blue, methyl red, and thymol blue³, with a pH reading range from 1 to 11.

Second, we confirmed the pH of PAA solutions with respect to the PAA concentration by using the pH indicator and also by using a pH electrode (Orion™ 8103 ROSS™, Thermo Scientific, USA), which revealed that the pH is about 2-3 (ranging from 3.55 for 0.1wt% PAA to 2.77 for 2.0wt% PAA). By correlating the color of the pH indicator and the pH data from the electrode, we identified the gray value representing pH (1.5-6) and PAA concentration (0-2.0 wt%) together (Fig. R2a). If we describe the local pH and PAA concentration profiles for a representative electroconvective viscous fingering, three regions are observed (Fig. R2b-c): i) diffusive ion enrichment zone on the top CEM with a linear increase of PAA concentration, ii) bulk region where the initial pH/concentration maintained (i.e., region (b) in Fig.1c), and iii) ion depletion zone on the bottom CEM with EC and corresponding flat pH/concentration profiles (i.e., region (a) in Fig.1c). During 8 sec operations at the applied voltage of 30 V in 0.5 wt% PAA solution, the diffusive enrichment zone expands only about 0.2 mm (which is well matched with the theoretical diffusion length, $0.1 \text{ mm} \sim \sqrt{4D_{PAA}t}$); whereas EC fingering grows much faster up to ~1 mm (Fig. R2c-d) (D_{PAA} is the diffusivity of PAA, see Supplementary Table 5). Interestingly, we can observe the EC-induced current hotspot between the fingers where the influx of the vortices occurs (white arrows in Fig. R2a)⁴. At this point, not only the depletion zone is suppressed, but also PAA is concentrated (Fig. R2d). Accordingly, as the viscosity between the fingers increases, downward flows between the fingers are considerably suppressed whereas there are upward flows in the fingers (Fig. R3). Such phenomenon is strengthened when EC are densely packed (e.g., 0.5 wt%, 30 V in Supplementary Video 1).

As described above, there are spatiotemporal variations of pH and PAA concentration in the ion enrichment /depletion zones during the developing EC fingers. However, we thought the scaling analysis was still valid due to the following three reasons. First, the bulk region completely separates the ion enrichment and depletion zones until EC touches the enrichment zone after a considerable time (e.g., > 14 sec at the applied voltage of 30 V in 0.5 wt% PAA solution, Fig. R2c-d); at this merging moment, EC already determine its shape. Also, while the viscosity between EC vortices increases as the PAA concentration increases, this is just one of the consequences of EC after it emerged. EC shape (circular or straight or ramified fingers) is still determined by the viscosity gradient of the ion depletion zone and the bulk region. In addition, we estimated shear rates from the particle tracking images, and confirmed that it is mostly below 1 s^{-1} (Fig. R3). Referring to Supplementary Fig. 12, the shear-thinning effect is not significant at below 1 s^{-1} shear rate, so we also can neglect the viscosity change by shear thinning of polyelectrolytes induced by EC vortices.

Lastly, the pH variation in whole regions is between 1.5 to 4.5 (Fig. R2c-d). Therefore, we can expect the PAA molecules to be in a globular form and the radius of gyration is nearly constant everywhere¹.

In the revised manuscript, we added the above new analysis and discussions as follows:

Manuscript page 18, line 420 – 429:

The anionic fluorescent dye (1.2 μM Alexa Fluor 488, Invitrogen, CA) or universal pH indicator (pH 1-11) (Hydrion One Drop, Micro Essential Laboratory Inc., USA) was dissolved in the solution to visualize EC. The pH indicator was composed of bromothymol blue, methyl red, and thymol blue³. We measured the pH of PAA solutions with respect to the PAA concentration by using the pH indicator and also by using a pH electrode (OrionTM 8103 ROSSTM, Thermo Scientific, USA), which revealed that the pH is about 2-3 (ranging from 3.55 for 0.1wt% PAA to 2.77 for 2.0wt% PAA). By correlating the color of the pH indicator and the pH data from the electrode, we identified the gray value representing pH (1.5-6) and PAA concentration (0-2.0 wt%) together (Supplementary Fig. 4).

Manuscript page 9, line 193 – 197:

It is noted that there are spatiotemporal variations of pH and PAA concentration in the ion enrichment /depletion zones during the developing EC fingers (Fig. 2 and Supplementary Fig. 4). While this variation may induce the changes of local rheological properties, the scaling analysis would be still valid because it does not affect to the boundary of the ion depletion zone and the bulk electrolyte when EC determine its shape (see Supplementary Note 4).

Manuscript page 18, line 431 – 433:

Since the shear rate in the experiment is low enough to neglect the shear thinning effect ($< 1 \text{ s}^{-1}$, see Supplementary Fig. 2), we use the zero-shear viscosity of the polyelectrolyte solutions.

Comment #3-2. *The ion transport (e.g. IV-curve) in the system is not characterized in the current manuscript. Are the observations all made in what is known as the overlimiting current regime, as this is where EC is induced? And how does the formation of these finger-like patterns influence the overall current? Is the current dropping as the fingers develop further? From the particle tracking images, it seems that EC is reduced as the fingers develop, which would indicate that the current goes down as no salt is replenished to the CEM. Authors should provide the current vs time data for characteristic experiments (e.g. the experiments shown in Supplementary figure 2, or main manuscript figure 2) so that a conclusion can be drawn on how this phenomenon actually influences ion transport.*

In follow up to this, on page 10 (line 224) the authors state that more EC will lead to more fingering. This in principle is true, but at the same time this will reduce the EC (e.g. the development of the fingers seems to suppress EC), is there a self-dampening effect beyond which this phenomenon becomes steady?

Response #3-2: We appreciate the reviewer's helpful comment. In the revised manuscript, we measured the current-voltage responses from 0 to 7 V at 0.2 V intervals every 30 s and the current-time responses in various PAA concentrations (0-2.0 wt%) (Fig. R4-5). Through this additional experiment, we first verified that three representative current regimes (Ohmic, limiting, and overlimiting) are clearly observed (Fig. R4a)⁶. Here, the critical onset voltage of EC initiating the overlimiting regime remains steady at ~3-4 V, regardless of whether EC shape is circular or straight/ramified fingers. As like with conventional EC, the current decreases exponentially in time as strong ion depletion zone and EC are developed⁶.

Next, from the current-voltage responses, we estimated the Ohmic and overlimiting conductance, which represents only ion conduction and electroconvective ion transport, respectively (Fig. R4b). As can be seen, both Ohmic and overlimiting conductance is keep nearly constant at < 1.0 wt%, but it starts to increase > 1.0 wt%. It is probably because considerable amount of PAA's counter ions (i.e., H⁺) is released as PAA concentration increases. Here, the ratio of these two conductance shows similar value at every PAA concentration (Fig. R4c); this indicates that the degree of electroconvective ion transport normalized by electromigration is not changed even PAA concentration increases up to 2.0 wt%.

Based on the analysis above, we identified that the fluid elasticity of the 0-2.0 wt% PAA solutions did not affect on EC in this work (please refer to the response #1-6).

Furthermore, we did not observe any effect of fingering on damping EC. When fingering occurs, as we discussed in the response #3-1, there are areas where the fingers grow (Fig. R2a, A-A') and hotspots where PAA accumulates between the fingers (Fig. R2a, B-B'). The hotspot becomes more viscous, which inhibits flow but does not inhibit flow where the finger grows. The particle tracking images in Fig. R3 show that the finger continues to grow even when the flow is inhibited in the hotspots. In conclusion, there is no self-dampening effect as the fingering develops over time.

Manuscript page 11, line 249 – 282:

Next, we investigate the effects of the fluid elasticity on the electroconvective viscous fingering in aspects of EC and viscous fingering (Fig. 4). According to Li *et al.*⁵, the fluid elasticity does not affect to the onset of EC, but it has two effects on EC: i) it reduces EC vortex strength and corresponding electroconvective ion transport, resulting a lower overlimiting conductance in current-voltage responses,

and ii) it destabilizes steady EC and promotes faster transition to chaotic EC at relatively low voltage. To investigate these effects of the fluid elasticity in our experiment, we measured the current-voltage responses and the current-time responses in various PAA concentrations (0-2.0 wt%) (Supplementary Fig. 6-7). In current-voltage responses with three representative current regimes (Ohmic, limiting, and overlimiting), we confirmed that the critical onset voltage of EC initiating the overlimiting regime remains steady at ~3-4 V, and the ratio of the overlimiting / Ohmic conductance is also keep nearly constant at every PAA concentrations, regardless of whether EC shape is circular or straight/ramified fingers (Supplementary Fig. 6c)⁶. This indicates that the degree of electroconvective ion transport normalized by electromigration is not changed even PAA concentration increases up to 2.0 wt%. As like with conventional EC, the currents also decrease exponentially in time as strong ion depletion zone and EC are developed⁶.

Manuscript page 22, line 484 – 488:

In the experiments of 0.5wt% and 1.0wt%, where the fingers appear densely, the PAA accumulates between the fingers as the inflow of ECs to the membrane makes a hotspot (see Supplementary Fig. 7). The hotspot becomes more viscous, which inhibits flow but does not inhibit flow where the finger grows. The particle tracking images in Supplementary Fig. 2 show that the finger continues to grow even when the flow is inhibited in the hotspots.

Comment #3-3. *Authors assume the diffusivity of Na⁺ and Cl⁻ to be constant independent of the viscosity of the surrounding fluid, while the diffusivity of PAA is considered to change. Can they show with an order of magnitude analysis that the assumption is reasonable?*

Response #3-3: We appreciate the reviewer's careful comment. In our scaling analysis, we defined the diffusivity in the anodic side of the CEM (i.e., the depletion zone, region (a)) and the bulk electrolyte (region (b)) (Fig. 1).

In the experiments with PQ-10, region (a) becomes concentrated with PQ-10 molecules. According to Ariel *et al.*¹³, if the size of the substance (e.g., ions) is much smaller than the intermolecular distance of the polymer in the solution, the substance barely interacts with the polymers, so the diffusivity of the substance is the same with that in the solvent without the polymer. In this analogy, PQ-10's radius of gyration (~223 nm) is much larger than the ionic radius of Na⁺ / Cl⁻ (~0.1 nm), and the intermolecular distance is in the order of the radius of gyration even in the high concentration¹⁴. Therefore, we can assume that the diffusivity of Na⁺ and Cl⁻ in the PQ-10 solution would be constant in our experiment.

In the experiments with PAA, since the PAA molecules migrate toward the anode, the depletion zone (region (a)) is assumed to be a pure water as most of PAA molecules are depleted. Therefore, we adopt the the diffusivity of Na⁺ and Cl⁻ in water, even the PAA's radius of gyration (~2.54 nm) is not much higher than the size of Na⁺ / Cl⁻, resulting $D_{Na^+} = 1.33 \times 10^{-9} \text{ m}^2/\text{s}$, $D_{Cl^-} = 2.03 \times 10^{-9} \text{ m}^2/\text{s}$, and $D_{eff} = 2/(1/D_{Na^+} + 1/D_{Cl^-}) = 1.6071 \times 10^{-9} \text{ m}^2/\text{s}$. In the original manuscript, we did not consider the effects of polyelectrolytes on ion diffusivity, and simply calculated the effective diffusivity by averaging the diffusivities of Na⁺, Cl⁻, and PAA. We corrected this content in the revised manuscript in Supplementary Note 1 and Supplementary Table 1.

Supplementary information page 3, line 58 – 73:

In PAA or PQ-10 solutions, the diffusivity of Na^+/Cl^- are presumed not changed in our experiment and scaling analysis. In the experiments with PQ-10, region (a) becomes concentrated with PQ-10 molecules. According to Ariel *et al.*¹³, if the size of the substance (e.g., ions) is much smaller than the intermolecular distance of the polymer in the solution, the substance barely interacts with the polymers, so the diffusivity of the substance is the same with that in the solvent without the polymer. In this analogy, PQ-10's radius of gyration (~ 223 nm) is much larger than the ionic radius of $\text{Na}^+ / \text{Cl}^-$ (~ 0.1 nm), and the intermolecular distance is in the order of the radius of gyration even in the high concentration¹⁴. Therefore, we can assume that the diffusivity of Na^+ and Cl^- in the PQ-10 solution would be constant in our experiment. In the experiments with PAA, since the PAA molecules migrate toward the anode, the depletion zone (region (a)) is assumed to be a pure water as most of PAA molecules are depleted. Therefore, we adopt the the diffusivity of Na^+ and Cl^- in water, even the PAA's radius of gyration (~ 2.54 nm) is not much higher than the size of $\text{Na}^+ / \text{Cl}^-$, resulting in $D_{\text{Na}^+} = 1.33 \times 10^{-9} \text{ m}^2/\text{s}$, $D_{\text{Cl}^-} = 2.03 \times 10^{-9} \text{ m}^2/\text{s}$, and $D_{\text{eff}} = 2/(1/D_{\text{Na}^+} + 1/D_{\text{Cl}^-}) = 1.6071 \times 10^{-9} \text{ m}^2/\text{s}$, where D_{Na^+} and D_{Cl^-} is the diffusivity of Na^+ and Cl^- , respectively. Viscosities and effective diffusivities of PAA and PQ-10 solutions are summarized in Supplementary Table 1.

Comment #3-4. *Can the authors comment on the physical relevance of the boundaries shown in Figure 3a? This figure seems to be of limited relevance in comparison to the other figures in the manuscript. My suggestion would be to further clarify why these boundaries are relevant (e.g. compare with traditional viscous fingering) or to combine these S_1 and S_2 dimensionless parameters with some influence of the applied electric field or the concentration of the polyelectrolyte solution.*

Response #3-5: We appreciate the reviewer's helpful comment. In previous papers, the shapes of traditional viscous fingering have been characterized through various methods, including roughness of interface¹⁵, fractal dimension¹⁶, and relative finger length to the total distance from the inlet¹⁷. While these methods are different from each other, they all successfully separate the blunt and fractured shapes of the finger.

In our investigation, we need to characterize three representative structures: circular EC vortex, straight finger, and ramified finger (Fig. 2g-i). The above methods, while effective in categorizing traditional viscous fingers, cannot capture these three structures because of the similar contour shapes of the circular EC and straight finger. Therefore, as addressed in the original manuscript, we adopted the simpler approach of using aspect and area ratios by separating individual fingers. The aspect ratio, S_1 , effectively distinguishes the straight and ramified finger since the latter appears wider with ramified branches for the same length (Fig. 2h, i). This S_1 cannot distinguish the circular EC and ramified finger, so we use the area ratio S_2 that can capture the compact shape of the circular EC (Fig. 2g, i).

In this work, we successfully distinguished the circular EC vortex and straight / ramified fingers with two shape factors (S_1 , S_2), and predicted their boundaries with two nondimensional parameters (M , $\sqrt{Ra_E}/(ScM)$) (Fig. 3). However, we did not investigate the relationship between these shape factors and control parameters here. Comprehensive studies for clarifying detailed morphological properties of EC fingers including wavelength, aspect ratio, and fractal dimension, are essential to elucidate this relationship. We mentioned the limit of the current research and potential future works in the revised manuscript as follows.

Manuscript page 6, lines 132 – 139:

The shapes of traditional viscous fingering have been characterized through various methods, including roughness of interface¹⁵, fractal dimension¹⁶, and relative finger length to the total distance from the inlet¹⁷. These methods, while effective in categorizing traditional viscous fingers, cannot distinguish circular EC and straight fingers due to their similar contour shapes. To distinguish three EC morphologies quantitatively, therefore, we define two geometric factors that represent the aspect and area ratios of each EC vortex: $S_1 = L/w$ and $S_2 = A_1/A_2$, where EC vortex width (w), length (L), the area occupied by each finger (A_1) and the area of the bounded rectangle of the finger (A_2) (Fig. 2(g)-(i)).

Manuscript page 11, lines 245 – 248:

In this work, we did not investigate the relationship between the shape factors (S_1 and S_2) and control parameters. A further comprehensive study is essential for clarifying detailed morphological properties of EC fingers including wavelength, aspect ratio, and fractal dimension to elucidate this relationship.

Comment #3-5. *The cross-references between the manuscript and the supplementary information should be checked. For example, on page 8, line 164, authors refer to Supp Text 3, which should be 2. There were a few more instances like this.*

Comment #3-6. *On page three, line 42, authors speak of local ion generation and consumption, but ions are not generated nor consumed. They are merely concentrated and depleted because they move in the electric field and are blocked by the ion exchange membrane of similar charge, or are selectively transported through the ion exchange membrane of opposite charge. This should be clarified prior to publication to avoid confusion.*

Response #3-5, 6: We apologize for the errors and inaccurate explanations. Upon careful review, we corrected the errors and unclear sentences.

Revised supplementary figures: Supplementary Figure 1

Figure R1 (Supplementary Fig. 1 in the revised supplementary information). The visualization device. a, Picture of the visualization device. **b**, Schematic of the device (cross-sectional view). The main channel for the visualization is between the cation exchange membranes (CEMs), and two side channels are between the CEM and the electrode. **c**, Example of the experimental image with applied voltage of 30 V in 1 wt% PAA solution (red rectangular region of a).

Added supplementary figures: Supplementary Figure 4:

Figure R2 (Supplementary Fig. 4 in the revised supplementary information). pH and PAA concentration profiles with electroconvective viscous fingering. **a**, Correlations between pH, PAA concentration and the gray value in microscopic images. The red line represents the correlation of the gray value with PAA concentration, and the black line represents the correlation of the gray value with pH. The X marks represent the experimental data, and the solid lines represent the best-fit lines. Here, we can identify the upper / lower limit of detections of PAA concentration as $> 2 \text{ wt}\%$ and $< 0.1 \text{ wt}\%$. **b**, The experimental image captured when the finger sufficiently develops (after 8 s since the voltage applied). **c-d**, pH and PAA concentration variations across the finger (A-A' in (b)) and between the finger (i.e., current hotspot, B-B' in (b)), respectively.

Revised supplementary figure: Supplementary Figure 2

Figure R3 (Supplementary Fig. 2 in the revised supplementary information). Velocity profile of the electroconvective viscous fingering at 30V in 1.0wt% PAA solution. **a**, Stacked images of electroconvective viscous fingering with 10 μm fluorescent particles (FluoSpheres™ polystyrene, Invitrogen, CA). The CEM is located at the bottom of the images. **b**, Schematic image of the flow field around the electroconvective viscous fingering. As EC in region (a) (white region) penetrates into region (b) (red region), electroconvective viscous fingering occurs and the fluid flows across the fingers in a vortex shape. The images were taken at a speed of 20 frames per second and stacked with 20 frames (1 s). Shear rate images are represented below the stacked images, which shows the shear rate remains below a nearly 1 s^{-1} . This demonstrates the shear-thinning effect is trivial to the growth of finger. We can observe the clear three streams of the fingers and vortex fields around them.

Added supplementary figure: Supplementary Figure 6

Figure R4 (Supplementary Fig. 6 in the revised supplementary information). The current-voltage responses and conductances in various PAA concentrations (0-2.0 wt%) **a**, Current-voltage curves according to various PAA concentrations (0-2.0 wt%). Ohmic, limiting, and overlimiting regimes are clearly observed in all cases. **b**, Ohmic conductance (σ_{Ohmic}) and overlimiting conductance ($\sigma_{\text{overlimiting}}$) and **c**, the ratio of them according to PAA concentrations. σ_{Ohmic} is the slope of current-voltage curve in the Ohmic regime, and $\sigma_{\text{Overlimiting}}$ is that in the overlimiting regime. By the release of PAA's counter ion (i.e., H^+), two conductance start to increase at > PAA 1.0 wt%.

Added supplementary figure: Supplementary Figure 7

Figure R5 (Supplementary Fig. 7 in the revised supplementary information). a-f, Current-time curves according to various PAA concentrations (0-2.0 wt%) and various applied voltages (1-50 V). Current response was measured for 60 s.

Added supplementary figure: Supplementary Figure 8

Figure R6 (Supplementary Fig. 8 in the revised supplementary information). The first normal stress difference (N_1) vs. shear rate curve of PAA solutions. The rheometer test was conducted with ARES-G2 Rheometer (TA instruments, New Castle, DE) in the shear rate range of 0.01-900 1/s. The averaged N_1 used in Fig. 4 in the manuscript is averaged in the shear rate range of $< 1 \text{ s}^{-1}$, which is depicted as a red arrow.

Added figures: Figure 4

Figure R7 (Fig. 4 in the revised manuscript). Fluid elasticity effects on electroconvective viscous fingering. **a**, The first normal stress difference (N_1), **b**, viscosity (μ_b), and **c**, yield stress (τ_y) vs. PAA concentrations. N_1 was averaged in shear rate range ($< 1 \text{ s}^{-1}$) because EC experience the shear rate lower than 1 s^{-1} (N_1 data are obtained from the rheometer tests in Supplementary Figure 8). **d**, The onset time of EC, **e**, The number of fingers and **f**, its width were measured from the experiment in Fig 2 and Supplementary Video 1 (the applied voltage of 30V in 0.1-2 wt% PAA solutions). $c^* = 0.84$ wt% is the overlap concentration of the PAA solution.

Revised supplementary figures: Supplementary Figure 5:

Figure R8 (Supplementary Fig. 5 in the revised manuscript). EC in various types of polyelectrolyte/polymer on AEM and CEM. a, Viscosities of the four polyelectrolyte solutions (neutral: polyethylene oxide (PEO), weak anionic: polyacrylic acid (PAA), strong anionic: sodium polystyrene sulfonate (NaPSS), weak cationic: polyallylamine hydrochloride (PAH), and strong cationic: Polyquaternium-10 (PQ-10)). The viscosities of the solutions were measured as 1.48, 1.7, 40, and 216 mPa s for 1wt% NaPSS, PAH, PAA, and 0.5wt% PQ-10 solution, respectively. The viscosity of PEO 1wt% solution is 2.2 mPa s, which is not displayed in graph. **b**, Microscopic images of EC in 10 mM NaCl solution and in the neutral polymer solution (1 wt% PEO). The circular EC vortices appear on both AEM and CEM. **c**, Microscopic images with anionic/cationic weak/strong polyelectrolytes. 1 wt% of PAA, NaPSS, PAH, and 0.5 wt% of PA-10 solutions were used. The polyelectrolytes suppress the EC on the side where they are concentrated under the electric field, while the neutral polymer does not at moderate concentration (1wt%). The electroconvective viscous fingering emerges in PAA and PQ-10 solutions ($M = 40$ and 243 , respectively), where the viscosity increases significantly with their concentration. However, the EC in PAH and NaPSS solutions ($M = 1.9$ and 1.66 , respectively) exhibits a circular EC shape.

Added supplementary note: Supplementary Note 2: Additional condition to generate EC fingering.

Here we describe the additional condition resulting from experiments with five polyelectrolytes in Supplementary Fig. 5: i) polyacrylic acid (PAA) as a weak anionic polyelectrolyte, ii) polyquaternium-10 (PQ-10) as a strong cationic polyelectrolyte, iii) sodium polystyrene sulfonate (NaPSS) as a strong anionic polyelectrolyte, iv) polyallylamine hydrochloride (PAH) as a weak cationic polyelectrolyte, and v) polyethylene oxide (PEO) as a neutral polymer.

Unlike the finger-like EC appearing in PAA and PQ-10 solutions, the conventional circular EC was observed in 1 wt% PAH and 1 wt% NaPSS solutions, both of which have a marginal viscosity ratio ($M=1.9$ and 1.66 , respectively). We may increase PAH / NaPSS concentrations to achieve a higher viscosity ratio. However, this requires significantly high weight of these polyelectrolytes because their viscosity increase effect is relatively small (Supplementary Fig. 5a). If we increase the concentration to get high bulk viscosity (10 wt% of PAH / NaPSS), the solution's conductance ($9.8 / 23.16$ mS for PAH / NaPSS 10wt% solution, respectively) is too high to initiate strong ion depletion zone and EC (EC occurs when the concentration of NaCl solution at the membrane surface is sufficiently low for initiating ion depletion zone, and most of EC experiments are conducted under NaCl 0.1 M, where the conductance is 7.3 mS)^{8,9}. Consequently, we can find one additional condition to generate EC fingering, i.e., the viscosity increase effect of a polyelectrolyte should be superior to the conductance increase effect according to its concentration.

Added supplementary note: Supplementary Note 4: pH/PAA concentration profiling.

The pH change (1.5-6) and the PAA concentration (0-2 wt%) are represented by the gray value (Supplementary Fig. 4a). The correlation between pH and gray value is illustrated in the methods section in the manuscript. There are three regions according to the pH/PAA concentration profiles: i) diffusive ion enrichment zone on the top CEM with a linear increase of PAA concentration, ii) bulk region where the initial pH/concentration maintained (i.e., region (b) in Fig.1c), and iii) ion depletion zone on the bottom CEM with EC and corresponding flat pH/concentration profiles (i.e., region (a) in Fig.1c). During 8 sec operations at the applied voltage of 30 V in 0.5 wt% PAA solution, the diffusive enrichment zone expands only about 0.2 mm (which is well matched with the theoretical diffusion length, $0.1 \text{ mm} \sim \sqrt{4D_{PAA}t}$, see Supplementary Fig. 4b): whereas EC fingering grows much faster up to $\sim 1 \text{ mm}$ (Supplementary Fig. 4c-d). Interestingly, we can observe the EC-induced current hotspot between the fingers where the influx of the vortices occurs (white arrows in Supplementary Fig. 4a)⁴. At this point, not only the depletion zone is suppressed, but also PAA is concentrated (Supplementary Fig. 4d). Accordingly, as the viscosity between the fingers increases, downward flows between the fingers are considerably suppressed whereas there are upward flows in the fingers. Such phenomenon is strengthened when EC are densely packed (e.g., 0.5 wt%, 30 V in Supplementary Video 1).

As described above, there are spatiotemporal variations of pH/PAA concentration in the ion enrichment/depletion zones during the developing EC fingers. However, these variations, except for change at the bulk-depletion region, are negligible for the following three reasons. First, the bulk region completely separates the ion enrichment and depletion zones until EC touches the enrichment zone after a considerable time (e.g., $> 14 \text{ sec}$ at the applied voltage of 30 V in 0.5 wt% PAA solution); at this merging moment, EC already determine its shape. Consequently, we can assume that the region (a) and (b) have constant viscosities. Also, while the viscosity between EC vortices increases as the PAA concentration increases, this is just one of the consequences of EC after it emerged. EC shape (circular or straight or ramified fingers) is still determined by the viscosity gradient of the ion depletion zone and the bulk region (i.e., viscosity ratio M). In addition, we estimated shear rates from the particle tracking images, and confirmed that it is mostly below 1 s^{-1} (Supplementary Fig. 2). Referring to Supplementary Fig. 12, the shear-thinning effect is not significant at below 1 s^{-1} shear rate, so we also can neglect the viscosity change by shear thinning of polyelectrolytes induced by EC vortices. Lastly, the pH variation in whole regions is between 1.5 to 4.5 (Supplementary Fig. 4). Therefore, we can expect the PAA molecules to be in a globular form and the radius of gyration is nearly constant everywhere¹.

References

1. Laguecir, A. *et al.* Size and pH effect on electrical and conformational behavior of poly (acrylic acid): Simulation and experiment. *European polymer journal* **42**, 1135-1144 (2006).
2. Curran, S., Hayes, R., Afacan, A., Williams, M. & Tanguy, P. Properties of carbopol solutions as models for yield-stress fluids. *J Food Sci* **67**, 176-180 (2002).
3. Ragoonanan, V. & Suryanarayanan, R. Ultrasonication as a potential tool to predict solute crystallization in freeze-concentrates. *Pharmaceutical research* **31**, 1512-1524 (2014).
4. Stockmeier, F. *et al.* Direct 3D observation and unraveling of electroconvection phenomena during concentration polarization at ion-exchange membranes. *J Membrane Sci* **640**, 119846 (2021).
5. Li, G., Archer, L.A. & Koch, D.L. Electroconvection in a viscoelastic electrolyte. *Phys Rev Lett* **122**, 124501 (2019).
6. Rubinstein, A. *et al.* Rubinstein. (*No Title*) (1961).
7. Jangir, P., Herale, A., Mohan, R. & Chokshi, P. Role of viscoelastic fluid rheology in miscible viscous fingering. *International Journal of Engineering Science* **179**, 103733 (2022).
8. Nikonenko, V.V. *et al.* Intensive current transfer in membrane systems: Modelling, mechanisms and application in electrodialysis. *Advances in colloid and interface science* **160**, 101-123 (2010).
9. Zabolotsky, V. *et al.* Coupled transport phenomena in overlimiting current electrodialysis. *Separation and purification technology* **14**, 255-267 (1998).
10. Belashova, E., Mikhaylin, S., Pismenskaya, N., Nikonenko, V. & Bazinet, L. Impact of cation-exchange membrane scaling nature on the electrochemical characteristics of membrane system. *Separation and Purification Technology* **189**, 441-448 (2017).
11. Kim, S.J., Wang, Y.-C., Lee, J.H., Jang, H. & Han, J. Concentration polarization and nonlinear electrokinetic flow near a nanofluidic channel. *Phys Rev Lett* **99**, 044501 (2007).
12. Dydek, E.V. *et al.* Overlimiting current in a microchannel. *Phys Rev Lett* **107**, 118301 (2011).
13. Michelman-Ribeiro, A., Horkay, F., Nossal, R. & Boukari, H. Probe diffusion in aqueous poly (vinyl alcohol) solutions studied by fluorescence correlation spectroscopy. *Biomacromolecules* **8**, 1595-1600 (2007).
14. Dobrynin, A.V., Jacobs, M. & Sayko, R. Scaling of polymer solutions as a quantitative tool. *Macromolecules* **54**, 2288-2295 (2021).
15. Gao, T., Mirzadeh, M., Bai, P., Conforti, K.M. & Bazant, M.Z. Active control of viscous fingering using electric fields. *Nat Commun* **10**, 4002 (2019).
16. Lemaire, E., Levitz, P., Daccord, G. & Van Damme, H. From viscous fingering to viscoelastic fracturing in colloidal fluids. *Phys Rev Lett* **67**, 2009 (1991).
17. Bischofberger, I., Ramachandran, R. & Nagel, S.R. Fingering versus stability in the limit of zero interfacial tension. *Nat Commun* **5**, 5265 (2014).

REVIEWERS' COMMENTS

Reviewer #2 (Remarks to the Author):

The authors have clearly addressed the questions and issues I proposed for review.

The figures are now clear and of higher quality. Their content and text now can be read smoothly compared with the original versions.

Regarding questions 5 and 6, I am very satisfied with the authors' response. I appreciate the inclusion of supplementary figures and additional measurements to complete the explanation. I do believe the answers are well-founded and well-crafted. With that said, I recommend this article for publication in Nature Communications.

Reviewer #3 (Remarks to the Author):

the authors have addressed my questions adequately

Reviewer #4 (Remarks to the Author):

The authors have addressed my concerns and improved the manuscript, warranting publication.